# Channel-independent function of UNC-9/Innexin in spatial arrangement of GABAergic synapses in *C. elegans*

**Ardalan Hendi[1,2], Long-Gang Niu[3], Andrew William Snow[4], Richard Ikegami[5], Zhao-Wen Wang[3], Kota Mizumoto[1,2,4,6]***

[1]Department of Zoology, University of British Columbia, Vancouver, Canada; [2]Life Sciences Institute, University of British Columbia, Vancouver, Canada; [3]Department of Neuroscience, University of Connecticut Health Center, Farmington, United States; [4]Graduate Program in Cell and Developmental Biology, University of British Columbia, Vancouver, Canada; [5]Janelia Research Campus, Howard Hughes Medical Institute, Ashburn, United States; [6]Djavad Mowafaghian Centre for Brain Health, University of British Columbia, Vancouver, Canada

**Abstract** Precise synaptic connection of neurons with their targets is essential for the proper functioning of the nervous system. A plethora of signaling pathways act in concert to mediate the precise spatial arrangement of synaptic connections. Here we show a novel role for a gap junction protein in controlling tiled synaptic arrangement in the GABAergic motor neurons in *Caenorhabditis elegans*, in which their axons and synapses overlap minimally with their neighboring neurons within the same class. We found that while EGL-20/Wnt controls axonal tiling, their presynaptic tiling is mediated by a gap junction protein UNC-9/Innexin, that is localized at the presynaptic tiling border between neighboring dorsal D-type GABAergic motor neurons. Strikingly, the gap junction channel activity of UNC-9 is dispensable for its function in controlling tiled presynaptic patterning. While gap junctions are crucial for the proper functioning of the nervous system as channels, our finding uncovered the novel channel-independent role of UNC-9 in synapse patterning.

## Editor's evaluation

This work provides a highly valued addition to our understanding of innexin gene function in the nervous system. The authors describe here potential functions in synapse tiling. The paper should be of interest to researchers with an interest in molecular mechanisms governing nervous system development.

## Introduction

Precise neuronal innervation and synaptic connections with appropriate target cells are essential for proper functioning of the nervous system. During development, neurons communicate with their neighboring neurons to define their innervation pattern. Neuronal tiling is one effect of such inter-neuronal communication observed in many neuron types, where neighboring neurons within the same class extend axons or dendrites in a non-overlapping manner (*Cameron and Rao, 2010*; *Grueber et al., 2002*; *Grueber and Sagasti, 2010*; *Grueber et al., 2003*; *Zipursky and Grueber, 2013*). Distinct types of neuronal tiling and their regulators have been reported in many neuronal classes across species. For example, in the visual system of *Drosophila*, L1 lamina neuron axons are arranged in columns in the medulla such that they only form synaptic connections within a single column in a

*For correspondence: mizumoto@zoology.ubc.ca

Competing interest: The authors declare that no competing interests exist.

non-redundant manner (**Millard et al., 2007**). Down syndrome cell adhesion molecule 2 (DSCAM2) mediates tiling of L1 lamina neuron axons through contact-dependent repulsive interactions between neighboring L1 neurons (**Millard et al., 2007**). Similarly, DSCAM serves as a homophilic repulsive signal to mediate self-avoidance and tiling in the mouse retinal amacrine cells (**Fuerst et al., 2009**; **Fuerst et al., 2008**). Axons of R7 photoreceptor neurons in the *Drosophila* visual system tile with those of neighboring R7 neurons through the TGFβ/activin signaling pathway (**Ting et al., 2007**). In *Drosophila*, the dendrites of neighboring class IV dendritic arborization neurons extend their dendrites in a non-overlapping manner with their neighboring neurons within the same class through Furry, Hippo, and Tricornered (**Emoto et al., 2004**; **Emoto et al., 2006**). However, due to the technical limitations in labeling two neighboring neurons within the same class, our knowledge of genetic mechanisms that underlie neuronal tiling remains limited.

Tiling also occurs at the level of synapses. In *Caenorhabditis elegans*, dorsal-anterior (DA) motor neurons form *en passant* cholinergic chemical synapses onto the dorsal body wall muscles in a way that each presynaptic domain from a single DA neuron does not overlap with those from the neighboring DA neurons (**White et al., 1986**). Previously, we showed that Semaphorin and Plexin-dependent inter-axonal interaction instructs presynaptic tiling between two posterior DA neurons by locally inhibiting synapse formation (**Mizumoto and Shen, 2013a**).

Neurons use various conserved signaling and cell adhesion molecules for precise spatial arrangement of chemical synapses (**Sanes and Yamagata, 2009**; **Yogev and Shen, 2014**). Mutations in these genes lead to an aberrant number of chemical synapses, which may underlie various neurodevelopmental and psychiatric disorders including autism spectrum disorder (ASD), schizophrenia, and bipolar disorder (**Guilmatre et al., 2009**; **Mitchell, 2011**; **Südhof, 2008**; **Tabuchi et al., 2007**; **Tang et al., 2014**; **Wen et al., 2014**). Several works showed that neurons use inhibitory cues to locally restrict synapse formation. For example, Sema3F and its receptors, Neurophilin-2 and PlexinA3, locally inhibit synapse formation in the proximal dendritic regions of cortical layer V pyramidal neurons (**Tran et al., 2009**). In *Drosophila*, Wnt4 secreted from the M13 muscles controls specificity of neuromuscular junctions by locally inhibiting synapse formation (**Inaki et al., 2007**). In *C. elegans,* two Wnts, LIN-44 and EGL-20 determine the topographic presynaptic arrangement of two posterior DA motor neurons, DA8 and DA9 (**Klassen and Shen, 2007**; **Mizumoto and Shen, 2013b**). UNC-6/Netrin and its receptor UNC-5/DCC is required to inhibit presynaptic assembly in the DA9 dendrite (**Poon et al., 2008**).

In addition to chemical synapses, neurons also form electrical synapses through gap junction channels that mediate electrical coupling between the cells. Gap junctions consist of tetra-membrane spanning proteins called connexins (Cx) in mammals and innexins (INX) in invertebrates (**Hall, 2017**). While there is no sequence similarity between them, Cx and INX monomers assemble into hexamers or octamers on neighboring cells that dock together to form gap junctions between neighboring cells, through which neurons exchange small molecules and ions (**Sánchez et al., 2019**). For simplicity, we will hereafter refer to chemical synapses as synapses and electrical synapses as gap junctions. In addition, mammals have another family of gap junction proteins called pannexin (PANX), which shares sequence similarity with INX. Unlike Cxs and INXs which form gap junction channels, PANXs only form hemichannels that mediate the exchange of small molecules and ions between the cytoplasm and extracellular space (**Deng et al., 2020**; **Michalski et al., 2020**). While gap junction proteins function primarily as channels, growing evidence supports channel-independent roles as cytoskeletal regulators. For example, human Cx43 and *Drosophila* INX2/3/4 control B lymphocyte and border cell migration, respectively, independent of their channel activities (**Falk et al., 2014**; **Machtaler et al., 2011**; **Miao et al., 2020**). Mammalian Cx43 and Cx26 mediate glial migration through a channel-independent adhesive role (**Elias et al., 2007**). However, the channel-independent roles of gap junction proteins in the nervous system are not well known. Interestingly, alterations in gap junction activity are associated with synaptopathies manifested by abnormal chemical synapse number and function (**Lapato and Tiwari-Woodruff, 2018**; **Swayne and Bennett, 2016**). For example, increased Cx43 expression is observed in prefrontal cortex of post-mortem brain tissue of ASD patients (**Fatemi et al., 2008**). A PANX1 mutation is found in patients with intellectual disabilities (**Shao et al., 2016**). Upregulation of Cx43 and PANX1 is also associated with Alzheimer's disease (**Giaume et al., 2019**). Functional and structural interactions between synapses and gap junctions have been observed in many aspects of neurodevelopment and function, yet the molecular mechanisms are largely unknown (**Pereda, 2014**).

Previous electron microscopy (EM) reconstruction of the *C. elegans* nervous system revealed axonal and dendritic tiling of the dorsal and ventral D-type GABAergic motor neurons (DDs and VDs) (*White et al., 1986*). Here we developed a system to stably label two neighboring DD motor neurons (DD5 and DD6) with fluorescent markers. Using this system, we show distinct regulation of axonal and presynaptic tiling by EGL-20/Wnt and UNC-9/INX, respectively. In *egl-20* mutants, axonal tiling between DD5 and DD6 was severely disrupted, while their presynaptic tiling was largely unaffected. We found that in the *egl-20* mutant background, loss of *unc-9*, which encodes an INX gap junction protein, causes ectopic synapse formation in the distal axon of DD5, resulting in disrupted presynaptic tiling. Strikingly, mutant UNC-9 proteins that form either putative, constitutively closed or open gap junction channels could still rescue the presynaptic patterning defect of *unc-9* mutants, indicating that UNC-9's gap junction channel activity is dispensable for its function in controlling presynaptic tiling. Our results reveal a novel channel-independent role for a gap junction protein in controlling synapse patterning. As UNC-9 gap junctions are formed at the DD5 and DD6 presynaptic tiling border, we propose that UNC-9 serves as a positional cue to define presynaptic domains.

## Results

### Tiled axonal, dendritic, and presynaptic patterning of the DD5 and DD6 neurons

Cell bodies of six DD class of GABAergic motor neurons (DD1–DD6) reside in the ventral nerve cord (*Figure 1A*; *White et al., 1986*). Each DD neuron extends a longer dendrite anteriorly and a shorter dendrite posteriorly, where they form postsynaptic dendritic spines. From the anterior dendrite, each DD neuron sends a commissure dorsally where it bifurcates to extend axons both anteriorly and posteriorly within the dorsal nerve cord. DD neurons form *en passant* synapses along their axons onto the dorsal body wall muscles and the VD motor neurons (*White et al., 1986*). Previous EM reconstruction has shown that the axons and dendrites of neighboring DD motor neurons have minimal overlap, where they form gap junction channels (*White et al., 1986*). As a result of tiled axonal and dendritic patterning, the domains of presynaptic and postsynaptic sites from each DD neuron do not overlap with those of the neighboring DD neurons, thereby achieving a tiled synaptic pattern.

To visualize neurites of the two neighboring DD neurons in live animals, we created a transgenic marker stain in which all DD neurons express the membrane-associated GFP (GFP::CAAX) under the DD neuron-specific promoter (P*flp-13*), and the membrane-associated mCherry (mCherry::CAAX) under the DD6 neuron-specific promoter (P*plx-2*) (see Materials and methods) (*Figure 1B*). The *flp-13* promoter activity in DD6 is often substantially weaker than in the rest of DD neurons (*Figure 1B*, personal communications with Michael Francis). This results in an increased color contrast between DD5, which expresses only GFP, and DD6, which expresses both mCherry and GFP. Using this marker strain, we observed minimal overlaps between the axons and the dendrites of DD5 and DD6 (*Figure 1A, B and G*, *Figure 1—figure supplement 1*), which confirmed the previous EM data (*White et al., 1986*). To visualize presynaptic tiling of DD5 and DD6, we generated another transgenic marker strain, in which all DD neurons express GFP::RAB-3 presynaptic vesicle marker under the *flp-13* promoter, and DD6 expresses mCherry::RAB-3 under the *plx-2* promoter. Consistent with the tiled axonal projection pattern of DD5 and DD6, their presynaptic patterning also exhibited tiled innervation (*Figure 1A, C and H*).

Together, we showed for the first time, that DD5 and DD6 have tiled axons, dendrites, and presynaptic patterning in live animals, which are in agreement with the previous EM reconstruction data (*White et al., 1986*).

### EGL-20/Wnt inhibits the outgrowth of DD5 posterior axon and dendrite

We next asked what controls axonal and dendritic tiling between DD5 and DD6. Previous studies showed that Wnt signaling acts as a negative regulator of neurite outgrowth, including for D-type motor neurons in *C. elegans* (*Maro et al., 2009*; *Onishi et al., 2014*; *Zou, 2004*). Among the five Wnt genes (*lin-44, egl-20, cwn-1, cwn-2, mom-2*) in *C. elegans*, *egl-20* is expressed in the cells around preanal ganglions (*Whangbo and Kenyon, 1999*), which are located near the DD5 and DD6 axonal and dendritic tiling border. In the loss-of-function mutants of *egl-20(n585)*, which carries a missense

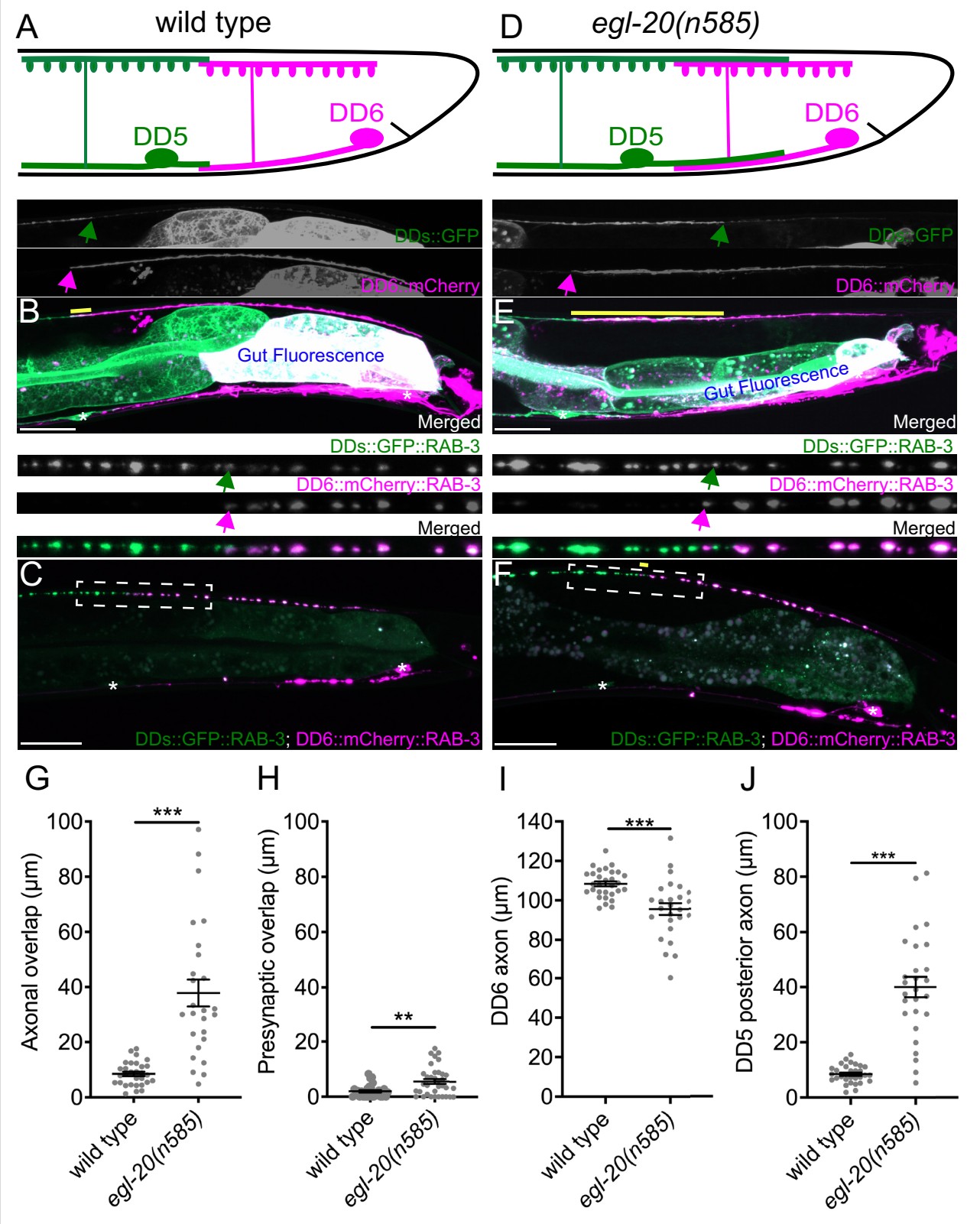

**Figure 1.** *egl-20/wnt* is required for axonal tiling between dorsal D-type 5 (DD5) and dorsal D-type 6 (DD6) neurons. (**A**) Schematic of axonal, dendritic, and presynaptic tiling between DD5 and DD6 in wild type. (**B**) Representative image of axonal tiling in wild type animals. Yellow line represents region of axonal overlap between DD5 and DD6. Green arrow indicates the end of DD5 posterior axon (top panel). Magenta arrow indicates the end of DD6 anterior axon (middle panel). (**C**) Representative image of presynaptic tiling in wild type animals. The magnified straightened image of the presynaptic

*Figure 1 continued on next page*

*Figure 1 continued*

tiling border, indicated by the dashed box, is shown above. Green arrow indicates the most posterior DD5 synapse (top panel). Magenta arrow indicates the most anterior DD6 synapse (middle panel). (**D**) Schematic of axonal and dendritic overlap between DD5 and DD6 of the *egl-20(n585)* mutant. (**E**) Representative image of axonal tiling in *egl-20(n585)* mutant animals. Yellow line represents region of axonal overlap between DD5 and DD6. Green arrow indicates the end of DD5 posterior axon (top panel). Magenta arrow indicates the end of DD6 anterior axon (middle panel). (**F**) Representative image of presynaptic tiling in the *egl-20(n585)* mutant. Yellow line represents region of presynaptic overlap between DD5 and DD6. The magnified straightened image of the presynaptic tiling border, indicated by the dashed box, is shown above. Green arrow indicates the most posterior DD5 synapse (top panel). Magenta arrow indicates the most anterior DD6 synapse (middle panel). Asterisks: DD5 and DD6 cell bodies. Scale bar: 20 μm. (**G**) Quantification of axonal overlap between DD5 and DD6. See *Figure 1—source data 1*. (**H**) Quantification of presynaptic overlap between DD5 and DD6. See *Figure 1—source data 2*. (**I**) Quantification of DD6 axonal length. See *Figure 1—source data 3*. (**J**) Quantification of DD5 posterior axonal length. See *Figure 1—source data 4*. See *Figure 1—figure supplement 1A* for the definition of the DD5 posterior axon. Each dot represents a single animal. See source data for sample size and individual value. Black bars indicate mean ± SEM. **$p<0.01$; ***$p<0.001$.

The online version of this article includes the following source data and figure supplement(s) for figure 1:

**Source data 1.** Quantification of axonal overlap between DD5 and DD6.

**Source data 2.** Quantification of overlap between DD5 and DD6 presynaptic domains.

**Source data 3.** Quantification of DD6 axon length.

**Source data 4.** Quantification of DD5 posterior axon length.

**Figure supplement 1.** Posterior overextension of dorsal D-type 5 (DD5) posterior dendrite and postsynaptic domain in the *egl-20(n585)* mutant.

**Figure supplement 1—source data 1.** Quantification of DD5 posterior dendrite length.

**Figure supplement 1—source data 2.** Quantification of DD5 postsynaptic domain length.

**Figure supplement 2.** The effects of *egl-20* and *unc-9* on axonal and dendritic length.

**Figure supplement 2—source data 1.** Quantification of axonal overlap between DD5 and DD6.

**Figure supplement 2—source data 2.** Quantification of DD6 axon length.

**Figure supplement 2—source data 3.** Quantification of DD5 posterior axon length.

**Figure supplement 2—source data 4.** Quantification of DD5 posterior dendrite length.

mutation in one of the conserved cysteine residues, we observed overextension of the DD5 posterior axon (*Figure 1D, E and J*), which resulted in significant overlap between DD5 and DD6 axons (*Figure 1G*). The length of DD6 axon was largely unaffected, or even slightly shorter in the *egl-20(n585)* mutant (*Figure 1I*), suggesting that the axonal tiling defect is a result of overextension of the DD5 posterior axon. Similar to the DD5 axon, the DD5 posterior dendrite was overextended posteriorly in the *egl-20(n585)* mutant (*Figure 1—figure supplement 1*). We could not reliably quantify the dendritic tiling defect due to the variable labeling of DD6 dendrite with mCherry::CAAX. Expression of *egl-20* from its endogenous promoter rescued the axonal overlap between DD5 and DD6 neurons, as well as the overextension of the DD5 posterior dendrite (*Figure 1—figure supplement 2*). From these observations, we conclude *egl-20* determines the length of the posterior axon and dendrite of DD5, thereby regulating the tiling of DD5 and DD6 axons.

## Axonal and presynaptic tiling are controlled by different mechanisms

Since DD5 and DD6 axons overlap in the *egl-20(n585)* mutant, we asked whether their presynaptic patterns are also compromised. If the presynaptic tiling between DD5 and DD6 is simply a consequence of axonal tiling, synapses will form throughout the DD5 axon in the *egl-20(n585)* mutants, resulting in significant overlap between the synaptic domains of DD5 and DD6. Surprisingly, despite the significant axonal overlap between DD5 and DD6, we observed little defect in the presynaptic tiling pattern (*Figure 1F and H*). While the degree of overlap between DD5 and DD6 presynaptic domains, which is defined as the distance between the most posterior DD5 presynaptic site and the most anterior DD6 presynaptic site, was significantly larger in the *egl-20(n585)* mutant compared with wild type, the small degree of presynaptic tiling defect did not reflect the large axonal overlap between DD5 and DD6 (*Figure 1G and H*). This observation strongly suggests that presynaptic tiling is not a consequence of axonal tiling, but rather there are additional mechanisms to tile their presynaptic domains even in the absence of axonal tiling.

As the DD5 posterior dendrite also overextends in the *egl-20(n585)* mutant, we next examined the postsynaptic dendritic spine patterning of DD5 using a transgenic strain that expresses ACR-12::GFP

under the *flp-13* promoter (*Philbrook et al., 2018*) (kind gift from M. Francis). ACR-12 is specifically localized at the postsynaptic sites of the GABAergic motor neurons (*Barbagallo et al., 2017*). Due to the weak expression from the *flp-13* promoter in DD6, ACR-12::GFP in DD6 is invisible in most animals, which allowed us to visualize the dendritic spines on the posterior dendrite of DD5. In wild type, the ACR-12::GFP puncta are localized throughout the length of DD5 dendrites (*Figure 1—figure supplement 1*). Consistently, the postsynaptic domain within the DD5 posterior dendrite, which we defined as the distance between DD5 cell body and the most posterior ACR-12::GFP punctum, is comparable to the length of DD5 posterior dendrite (*Figure 1—figure supplement 1*). In the *egl-20(n585)* mutant, the postsynaptic domain within the DD5 posterior dendrite is significantly longer compared with wild type and is comparable to the length of the DD5 posterior dendrite (*Figure 1—figure supplement 1*). Indeed, we observed that the ACR-12::GFP puncta are distributed throughout the length of the posterior DD5 dendrite (*Figure 1—figure supplement 1*). We could not examine the dendritic spine patterning of DD6 due to the weak expression of ACR-12::mCherry from the *plx-2* promoter. However, our results suggest that, unlike presynaptic tiling which is largely unaffected by the DD5 axon overextension in the *egl-20(n585)* mutant, the length of the DD5 postsynaptic domain is determined by the length of DD5 posterior dendrite.

## UNC-9/INX is required for presynaptic tiling between DD5 and DD6

We next sought to identify genes that are responsible for presynaptic tiling between DD5 and DD6 neurons. Previously we showed that Semaphorin-Plexin signaling mediated by PLX-1/Plexin, RAP-2/Rap2A, and MIG-15/TNIK is crucial for the presynaptic tiling of two DA-class cholinergic motor neurons (*Chen et al., 2018*; *Mizumoto and Shen, 2013a*). However, we did not observe a presynaptic tiling defect between DD5 and DD6 neurons in the double mutants of *plx-1(nc36); egl-20(n585)*, *egl-20(n585); rap-2(gk11)*, and *egl-20(n585); mig-15(rh148)* (*Figure 2—figure supplement 1*), suggesting that the presynaptic tiling between DD5 and DD6 is controlled by another mechanism. Previous EM studies have shown that gap junctions consisting of INX proteins are formed in the regions of minimal axonal and dendritic overlap between DD neurons where presynaptic and postsynaptic tiling are established (*White et al., 1976*; *White et al., 1986*). These gap junctions play crucial roles in electrical coupling between DD neurons during sinusoidal locomotion (*Kawano et al., 2011*). Given that the position of these gap junctions coincides with the presynaptic tiling border between DD5 and DD6, we tested their potential functions in presynaptic tiling. DD neurons express at least six INX genes, *inx-1, inx-2, inx-10, inx-14, unc-7,* and *unc-9* (*Altun et al., 2009*; *Spencer et al., 2011*; *Taylor et al., 2021*). Among these six innexins, double mutants of *egl-20(n585)* and *unc-9(e101* or *tm5479)* null mutants exhibited a significant presynaptic tiling defect (*Figure 2B and E*). Ectopic DD5 presynaptic puncta are formed in the DD5 posterior axon that overextended into the DD6 axonal region, which creates intermingled patterning of DD5 and DD6 presynaptic puncta (*Figure 2B*). The presynaptic tiling defect in the *egl-20(n585); unc-9(e101)* mutants is fully rescued by the P*egl-20::egl-20* transgene, which rescues the axonal tiling defect (*Figure 1—figure supplement 1*), suggesting that overextension of DD5 posterior axon is necessary for observing the effect of *unc-9* in presynaptic patterning (*Figure 2E*). Consistently, *unc-9(e101)* single mutants did not exhibit presynaptic tiling defect due to the normal axonal tiling in the presence of *egl-20* (*Figure 1—figure supplement 1*). Importantly, loss of *unc-9* does not enhance the axonal tiling defect of *egl-20(n585)* mutants (*Figure 1—figure supplement 1*). Therefore, the presynaptic tiling defect in the *egl-20(n585); unc-9(e101)* double mutants is not due to an increased axonal tiling defect.

To determine if the ectopic RAB-3 puncta present in the DD5 posterior axon of *egl-20(n585); unc-9(e101)* mutants represent bona fide synapses, we examined co-localization between mCherry::RAB-3 puncta in the DD5 neuron and NLG-1/Neuroligin, which is localized at the postsynaptic site of the GABAergic neuromuscular junctions (*Maro et al., 2015*; *McDiarmid et al., 2020*; *Tu et al., 2015*), using endogenously tagged NLG-1::AID::GFP (*McDiarmid et al., 2020*). We observed that the mCherry::RAB-3 puncta at the posterior DD5 axon of *egl-20(n585); unc-9(e101)* mutants are apposed by NLG-1::AID::GFP, similar to those in wild type (*Figure 2C and D*). The most posterior portion of the DD5 axon did not contain RAB-3 puncta apposed by NLG-1::AID::GFP (*Figure 2—figure supplement 2*). This is consistent with the fact that the degree of presynaptic tiling defect is much smaller than that of axonal tiling defect (compare *Figures 1G and 2E*) (see Discussion). The RAB-3 puncta in the DD5 axon are co-localized with the presynaptic active zone protein UNC-10 in the *egl-20(n585);*

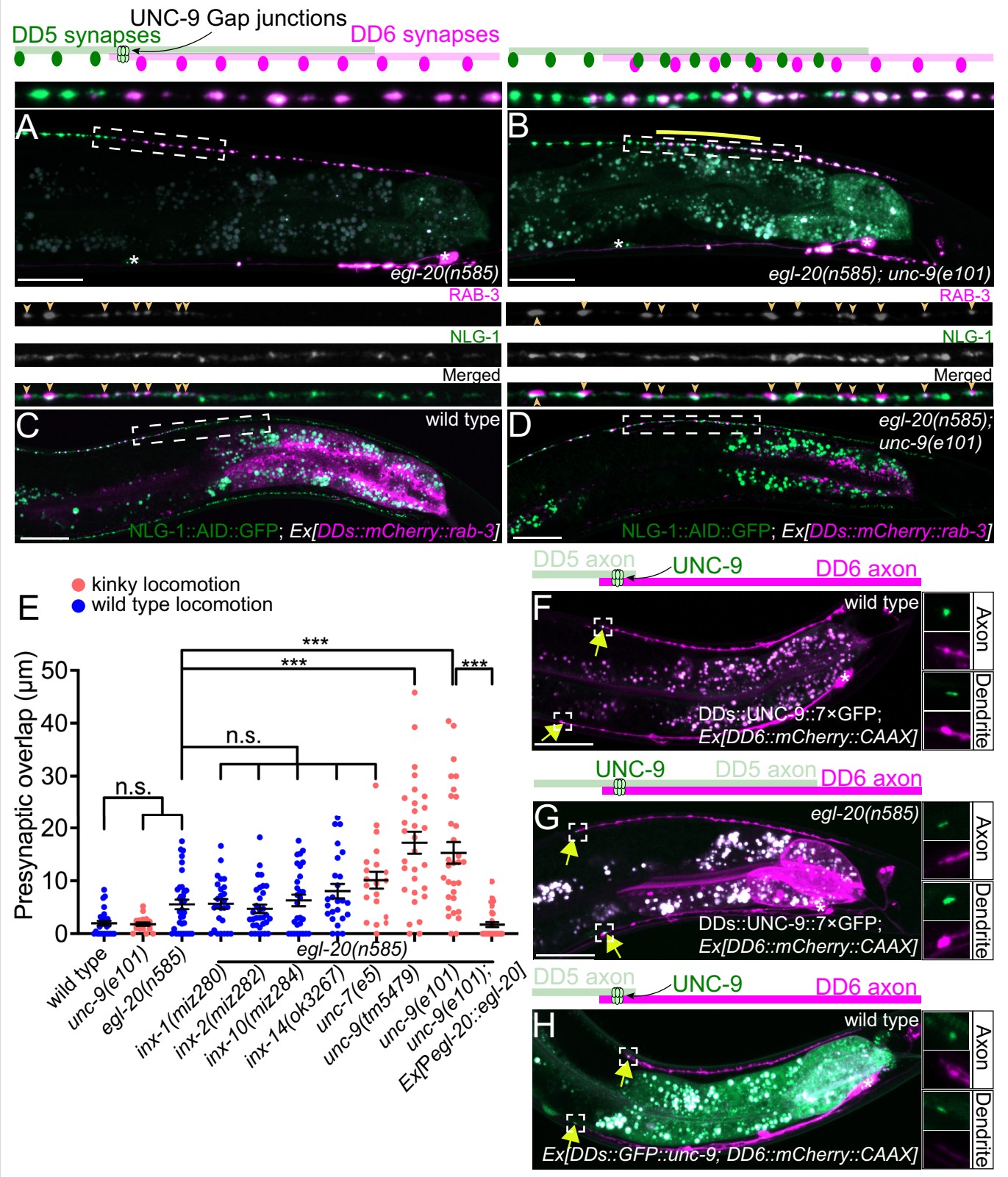

**Figure 2.** UNC-9/INX is localized at the presynaptic tiling border and is required for presynaptic tiling between dorsal D-type 5 (DD5) and dorsal D-type 6 (DD6) neurons. (**A–B**) Representative images of presynaptic tiling in the *egl-20(n585)* (**A**), and *egl-20(n585); unc-9(e101)* (**B**) mutants. The magnified images of the presynaptic tiling border and their schematics (indicated by dashed box) are shown above. Yellow line indicates region of presynaptic overlap between DD5 and DD6. (**C–D**) Representative images of mCherry::RAB-3 expressed in DD5 and endogenous NLG-1::AID::GFP in

*Figure 2 continued on next page*

*Figure 2 continued*

wild type (**C**) and *egl-20(n585); unc-9(e101)* double mutant (**D**). Dashed boxes indicate the region straightened for mCherry::RAB-3 (top panels), NLG-1::AID::GFP (middle panels), and merged channels (bottom panels) shown above. Yellow arrowheads indicate mCherry::RAB-3 puncta. (**E**) Quantification of presynaptic overlap between DD5 and DD6. See *Figure 2—source data 1*. Each dot represents a single animal. See source data for sample size and individual value. Black bars indicate mean ± SEM. n.s.: not significant; ***p<0.001. (**F–G**) Representative image of UNC-9::7×GFP localization at the anterior tip of the DD6 axon and dendrite (indicated by yellow arrows) in wild type (**F**) and *egl-20(n585)* mutant (**G**). (**H**) Representative image of GFP::UNC-9 localization at the anterior tip of the DD6 axon and dendrite (indicated by yellow arrows) in wild type. The magnified UNC-9::7×GFP (**F**, **G**) and GFP::UNC-9 (**H**) puncta and mCherry::CAAX signals in the anterior tip of DD6 axon and dendrite, indicated by the dashed boxes, are shown to the right of the merged images. Asterisks: DD5 and DD6 cell bodies. Scale bar: 20 μm.

The online version of this article includes the following source data and figure supplement(s) for figure 2:

**Source data 1.** Quantification of overlap between DD5 and DD6 presynaptic domains.

**Figure supplement 1.** Quantification of the presynaptic tiling defect between dorsal D-type 5 and dorsal D-type 6 in the mutants of known gap junction effectors and presynaptic tiling regulators.

**Figure supplement 1—source data 1.** Quantification of overlap between DD5 and DD6 presynaptic domains.

**Figure supplement 2.** RAB-3 puncta in the dorsal D-type 5 (DD5) axon represent bona fide synapses.

**Figure supplement 3.** Axonal and presynaptic tiling defects at the L2 stage.

**Figure supplement 3—source data 1.** Quantification of axonal overlap between DD5 and DD6.

**Figure supplement 3—source data 2.** Quantification of overlap between DD5 and DD6 presynaptic domains.

**Figure supplement 4.** Heat-shock-induced expression of *unc-9* at L2 stage did not rescue presynaptic tiling defect of *egl-20(n585); unc-9(e101)*.

**Figure supplement 4—source data 1.** Quantification of overlap between DD5 and DD6 presynaptic domains.

**Figure supplement 5.** UNC-9/INX localization at the anterior tip of the dorsal D-type 6 (DD6) dendrite and axon in L1 animals.

**Figure supplement 6.** UNC-9/INX localization in the mutants of *egl-20(n585)*, *zoo-1(tm4133)*, *nlr-1(gk366849)*, and *unc-104(e1265)*.

**Figure supplement 6—source data 1.** Quantification of UNC-9::7×GFP signal intensity at the tip of DD6 axon.

**Figure supplement 7.** *unc-9* mutant does not exhibit presynaptic tiling defect between dorsal-anterior 8 (DA8) and dorsal-anterior 9 (DA9) cholinergic motor neurons.

**Figure supplement 7—source data 1.** Quantification of overlap between DA8 and DA9 presynaptic domains.

*unc-9(e101)* double mutant (*Figure 2—figure supplement 2*). These observations indicate that the ectopic RAB-3 puncta formed in the posterior DD5 axon of *egl-20(n585); unc-9(e101)* double mutant likely represent bona fide synapses.

DD neurons undergo remodeling at the end of the first larval (L1) stage, when the dorsal neurites switch their fate from dendrite to axon (*White et al., 1978*). We found that the overgrowth of the DD5 posterior axon in *egl-20(n585)* mutant is present at the second larval (L2) stage (*Figure 2—figure supplement 3*). We observed a significant presynaptic tiling defect at the L2 stage in *egl-20(n585); unc-9(e101)* mutant (*Figure 2—figure supplement 3*). As the remodeling completes by the end of L2 stage, this result suggests that *unc-9* is required for the establishment of presynaptic tiling.

To determine if *unc-9* is also required for the maintenance of presynaptic tiling, we conducted rescue experiments using a heat shock promoter. We found that the heat-shock-induced expression of *unc-9* at the L2 stage after presynaptic tiling is established did not rescue the presynaptic tiling defects in *egl-20(n585); unc-9(e101)*. This result is consistent with the idea that *unc-9* is required for the establishment of the presynaptic tiling of DD5 and DD6 (*Figure 2—figure supplement 4*).

## UNC-9 is localized at the presynaptic tiling border between DD5 and DD6 axons

We next examined the subcellular localization of UNC-9 in DD neurons by labeling the endogenous UNC-9 using the split-GFP-based native and tissue-specific fluorescence method (*He et al., 2019*). Briefly, we inserted seven tandem repeats of last β-strand of GFP (*7×gfp11*) at the C-terminus of endogenous locus of *unc-9* using CRISPR/Cas9 genome editing. The *unc-9(miz81 [unc-9::7×gfp11])* animals exhibit uncoordinated locomotion pattern, probably because UNC-9::7×GFP11 is expected to form constitutively open gap junction channels (see below). We then expressed the remaining part of GFP (GFP1-10) specifically in DD neurons using the *flp-13* promoter to reconstitute the fluorescent UNC-9::7×GFP exclusively in the DD neurons. In wild type animals, UNC-9::7×GFP is localized at the anterior tip of the DD6 axon and dendrite (*Figure 2F*). Consistent with the requirement of *unc-9* in

the establishment of presynaptic tiling, UNC-9::7×GFP localization at the anterior tip of the DD6 axon and dendrite is observed in L1 animals (*Figure 2—figure supplement 5*). In the *egl-20(n585)* mutant, in which DD5 axon overextends to the DD6 axonal region, UNC-9::7×GFP was localized at the anterior tip of DD6 axon and dendrite similar to wild type, even though the signal intensity of the UNC-9::7×GFP at the tip of DD6 axon was slightly reduced (*Figure 2G* and *Figure 2—figure supplement 6*). The slight reduction of UNC-9 localization at the presynaptic tiling border may account for the subtle presynaptic tiling defect of the *egl-20(n585)* single mutant (*Figure 1H*). This result suggests that overextension of DD5 axon and dendrite in the *egl-20* mutant does not affect the overall position of UNC-9 gap junctions formed between DD5 and DD6 axons and dendrites. As presynaptic tiling was largely unaffected in *egl-20(n585)* single mutant, the UNC-9 localization at the presynaptic tiling border in *egl-20(n585)* mutant suggests that UNC-9 functions as a positional cue to define the presynaptic tiling border between DD5 and DD6 in *egl-20* mutant background. While the UNC-9::7×GFP localization appears to be unaffected in the dendrite, the DD5 posterior postsynaptic domain, judged by ACR-12::GFP, is significantly longer in the *egl-20(n585)* single mutant (*Figure 2G*, *Figure 1—figure supplement 1*), suggesting that *unc-9* does not control postsynaptic patterning of DD5.

To confirm that UNC-9::7×GFP represents the localization of endogenous UNC-9, we examined the localization of transgenically expressed N-terminally tagged GFP::UNC-9, which was shown to be functional (*Meng et al., 2016*). We found that GFP::UNC-9 expressed under the DD-neuron-specific promoter, *flp-13*, was localized to the anterior tip of the DD6 axon and dendrite, similar to UNC-9::7×GFP (*Figure 2H*).

The UNC-9::7×GFP localization at the presynaptic tiling border is reminiscent of the PLX-1::GFP localization at the presynaptic tiling border of DA8 and DA9 cholinergic motor neurons, where PLX-1 locally inhibits synapse formation in DA9 (*Mizumoto and Shen, 2013a*). We therefore examined the localization of UNC-9::7×GFP in the DA9 neuron by expressing GFP1-10 under the DA9-neuron-specific promoter, *itr-1*. Unlike in DD neurons where UNC-9::7×GFP is almost exclusively detected at axonal and dendritic tiling borders between neighboring DD neurons, UNC-9::7×GFP puncta was distributed throughout the DA9 axon (*Figure 2—figure supplement 7*). The UNC-9::7×GFP puncta are often associated with the presynaptic varicosity which is reminiscent to the perisynaptic localization of endogenous UNC-9 in DD neurons (*Yeh et al., 2009*). In addition, UNC-9::7×GFP was consistently localized at the distal end of DA9 dendrites. We also found that *unc-9(e101)* mutant did not exhibit a presynaptic tiling defect between DA8 and DA9 (*Figure 2—figure supplement 7*). While we do not exclude the possibility that other innexins control presynaptic tiling of DA8 and DA9 neurons, our result suggests that each motor neuron class uses distinct mechanisms for controlling the presynaptic tiling pattern.

Recent work in *C. elegans* and zebrafish showed that gap junction clustering is controlled by a neurexin-like receptor 1 (NLR-1) and a tight junction associated protein ZO-1 (*Meng and Yan, 2020*; *Lasseigne et al., 2021*). In the nerve ring of *C. elegans*, clustered UNC-9 localization depends on NLR-1 (*Meng and Yan, 2020*). In zebrafish, Cx35.5 and Cx34.1 localization at the club ending synapses depend on ZO1b (*Lasseigne et al., 2021*). While both *nlr-1* and *zoo-1*, an ortholog of *ZO-1*, are expressed in DD neurons (*Taylor et al., 2021*), the localization of UNC-9::7×GFP at the anterior tip of the DD6 axon was unaffected in the *nlr-1(gk366849)* and *zoo-1(tm4133)* null mutants (*Figure 2—figure supplement 6*), suggesting that UNC-9 localization at the axonal tiling border is independent of the known regulators of gap junction clustering. We note that UNC-9::7×GFP puncta at the DD6 dendrite had a tendency to localize slightly posteriorly from the tip of the DD6 dendrite in *zoo-1(tm4133)* mutant (*Figure 2—figure supplement 6*). Consistent with the unaffected UNC-9 localization, we did not observe a presynaptic tiling defect in the *egl-20(n585); nlr-1(gk366849)* and *egl-20(n585); zoo-1(tm4133)* mutants (*Figure 2—figure supplement 1*). We also examined UNC-9::7×GFP localization in the mutant of neuronal kinesin/KIF1A, *unc-104(e1265)*, in which axonal transport is severely compromised (*Hall and Hedgecock, 1991*). We found a slight reduction in UNC-9::7×GFP signal intensity without affecting the overall localization pattern at the anterior tip of DD6 axon (*Figure 2—figure supplement 6*). These results suggest that UNC-9 localization at the axonal tiling border is mediated by an unknown mechanism, and its transport may partially depend on UNC-104/KIF1A. We could not examine presynaptic tiling in *unc-104(e1265)* mutant, due to a severe defect in axonal transport of presynaptic materials.

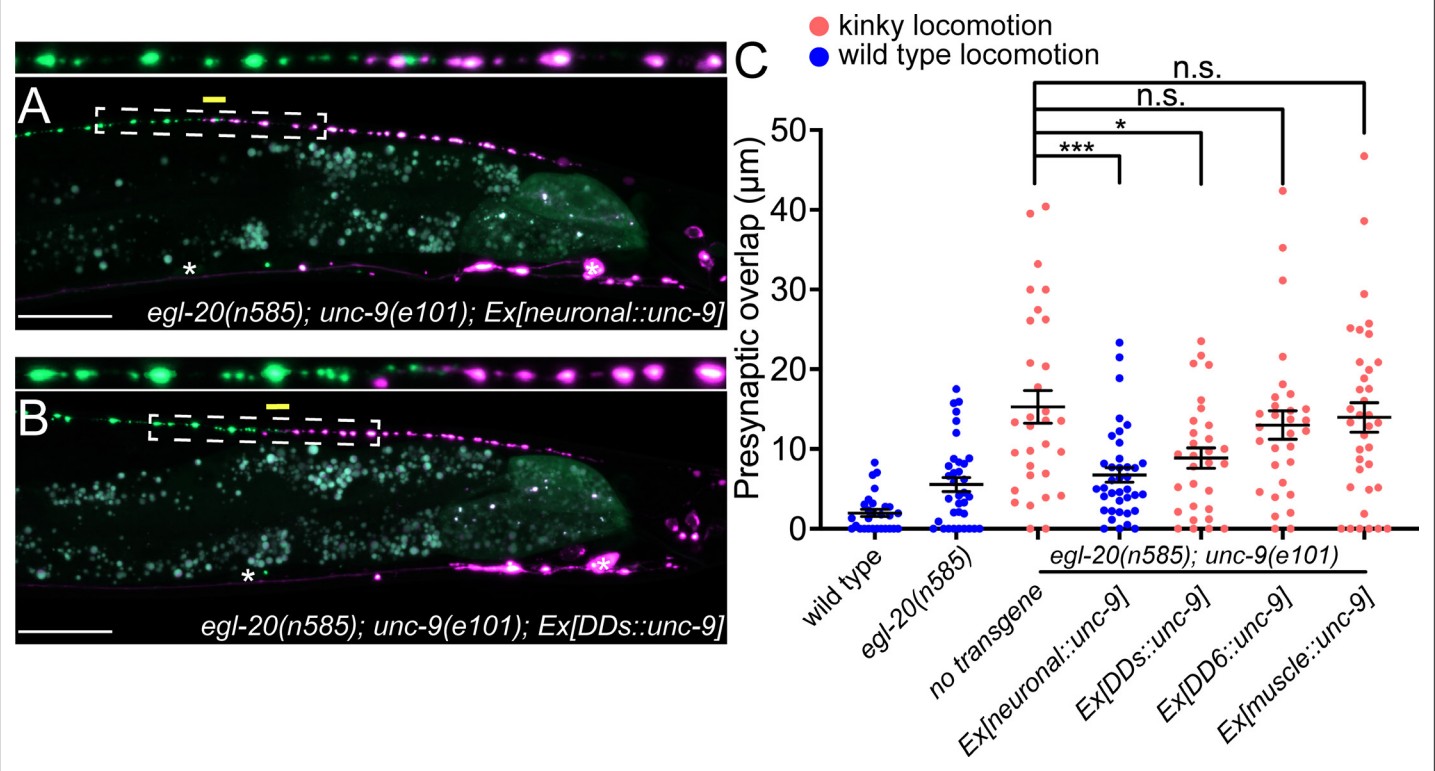

**Figure 3.** UNC-9/INX functions cell-autonomously to control presynaptic tiling between dorsal D-type 5 (DD5) and dorsal D-type 6 (DD6) neurons. (**A–B**) Representative images of presynaptic tiling in the *egl-20(n585); unc-9(e101)* double mutants with pan-neuronal expression of *unc-9* from the *rgef-1* promoter (**A**), and DD-neuron-specific expression of *unc-9* from the *flp-13* promoter (**B**). The magnified straightened image of the presynaptic tiling border, indicated by the dashed box, is shown above. Yellow lines indicate region of presynaptic overlap between DD5 and DD6. Asterisks: DD5 and DD6 cell bodies. Scale bar: 20 µm. (**C**) Quantification of presynaptic overlap between DD5 and DD6. Tissue-specific promoters used in these experiments are *rgef-1* (pan-neuronal), *flp-13* (DDs), *plx-2* (DD6), *myo-3* (body wall muscles). See *Figure 3—source data 1*. Each dot represents a single animal. See source data for sample size and individual value. Black bars indicate mean ± SEM. n.s.: not significant; *p<0.05; ***p<0.001.

The online version of this article includes the following source data for figure 3:

**Source data 1.** Quantification of overlap between DD5 and DD6 presynaptic domains.

## UNC-9 functions cell-autonomously in the DD neurons to control presynaptic tiling

*unc-9* is expressed broadly in the nervous system and non-neuronal tissues, including the DD neurons and their postsynaptic body wall muscles (*Yeh et al., 2009*). We therefore asked in which cells UNC-9 functions to control presynaptic tiling between DD5 and DD6 by conducting tissue-specific rescue experiments. Pan-neuronal expression of *unc-9* under the *rgef-1* promoter rescued the presynaptic tiling defect of the *egl-20(n585); unc-9(e101)* double mutants (*Figure 3A and C*). Similarly, *unc-9* expression in the presynaptic DD neurons using the *flp-13* promoter but not in the postsynaptic body wall muscles using the *myo-3* promoter rescued the presynaptic tiling defect (*Figure 3B and C*). These results suggest that *unc-9* controls tiled presynaptic patterning in the presynaptic DD neurons. The slightly weaker rescue activity of P*flp-13::unc-9* compared with P*rgef-1::unc-9* may be due to a weaker expression of *unc-9* from the *flp-13* promoter in the DD6 neuron. Expression of *unc-9* specifically in the DD6 neuron using the *plx-2* promoter did not rescue the presynaptic tiling defect between DD5 and DD6 (*Figure 3C*). These results suggest that *unc-9* is required in both DD5 and DD6 neurons to control tiled presynaptic patterning between DD5 and DD6.

## *unc-1/stomatin* is not required for the presynaptic tiling between DD5 and DD6

As UNC-9 forms gap junctions between DD5 and DD6, we next tested if UNC-9 controls tiled presynaptic patterning via its gap junction channel activity. It has been shown that UNC-1/stomatin

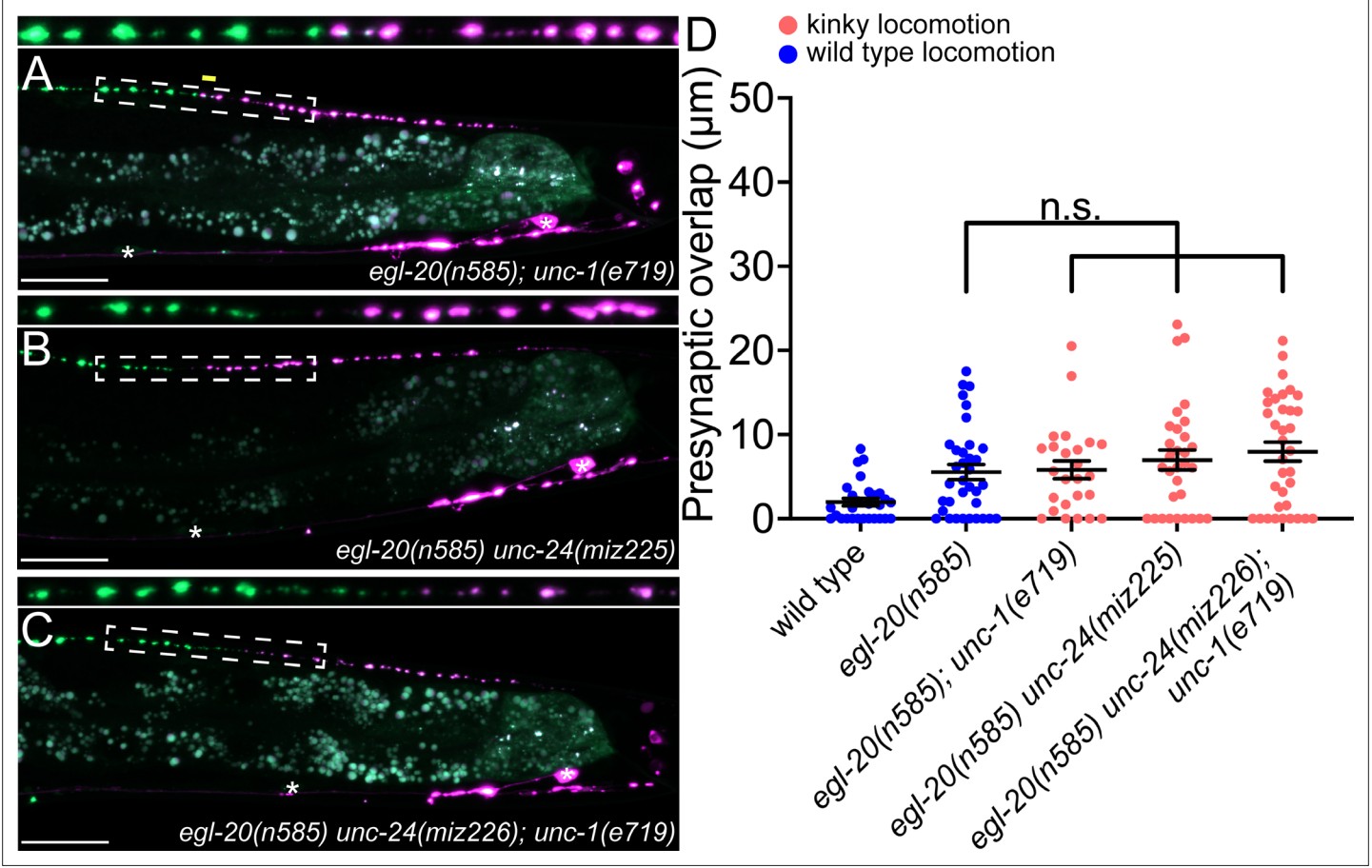

**Figure 4.** *unc-1/stomatin* is not required for presynaptic tiling between dorsal D-type 5 (DD5) and dorsal D-type 6 (DD6) neurons. (**A–B**) Representative images of presynaptic tiling in the *egl-20(n585); unc-1(e719)* (**A**), *egl-20(n585) unc-24(miz225)* (**B**), and *egl-20(n585) unc-24(miz226); unc-1(e719)* (**C**) mutants. The magnified straightened images of the presynaptic tiling border, indicated by the dashed box, are shown above. Yellow line indicates region of presynaptic overlap between DD5 and DD6. Asterisks: DD5 and DD6 cell bodies. Scale bar: 20 μm. (**D**) Quantification of presynaptic overlap between DD5 and DD6. See *Figure 4—source data 1*. Each dot represents a single animal. See source data for sample size and individual value. Black bars indicate mean ± SEM. n.s.: not significant.

The online version of this article includes the following source data and figure supplement(s) for figure 4:

**Source data 1.** Quantification of overlap between DD5 and DD6 presynaptic domains.

**Figure supplement 1.** Locomotion defects of UNC-9-gap junction channel defective mutants.

is essential for opening the UNC-9 gap junction channels (*Chen et al., 2007*; *Jang et al., 2017*). In the loss-of-function mutants of *unc-1*, UNC-9 gap junction channel activity is completely abolished (*Chen et al., 2007*). Due to the defective UNC-9 gap junction channel activity, *unc-1* mutants exhibit a kinky locomotion (kinker) phenotype similar to *unc-9* mutants (*Figure 4—figure supplement 1*). To test whether UNC-9's function in regulating presynaptic tiling depends on its gap junction channel activity, we examined the presynaptic patterning of DD5 and DD6 in the *egl-20(n585); unc-1(e719)* double mutants. Interestingly, unlike *egl-20(n585); unc-9(e101)* double mutants, *egl-20(n585); unc-1(e719)* double mutants did not exhibit a presynaptic tiling defect (*Figure 4A and D*). While UNC-9 gap junction activity is completely abolished in *unc-1* mutants, it is possible that another stomatin can mediate the UNC-9 gap junction activity in the DD neurons. We therefore examined *unc-24/stomatin*, as *unc-24* mutants also exhibit the kinker phenotype similar to *unc-1* and *unc-9* mutants (*Figure 4—figure supplement 1*). Similar to *egl-20(n585); unc-1(e719)* mutants, *egl-20(n585) unc-24(miz225)* double mutants and *egl-20(n585) unc-24(miz226); unc-1(e719)* triple mutants did not exhibit a presynaptic tiling defect (*Figure 4B, C and D*). These results raise the possibility that UNC-9's function in controlling presynaptic tiling between DD5 and DD6 is independent of its gap junction channel activity.

# A putative, constitutively closed form of UNC-9 can control presynaptic tiling

While Cxs and INXs function primarily as gap junction channels, recent studies have revealed channel-independent roles for gap junction proteins (*Dbouk et al., 2009*; *Elias et al., 2007*; *Kameritsch et al., 2013*; *Kameritsch et al., 2012*; *Miao et al., 2020*; *Olk et al., 2009*). For example, *Drosophila* INXs (Inx2/3/4) control border cell migration independent of their gap junction channel activity (*Miao et al., 2020*). Cx43 and Cx26 mediate the migration of radial glia cells in a channel-independent manner (*Elias et al., 2007*). Given that *unc-1* is dispensable for *unc-9's* function in presynaptic tiling, we tested if UNC-9 controls presynaptic tiling independent of its gap junction channel activity by using mutant UNC-9 with defective channel activity. Previous cryo-EM studies using *C. elegans* INX-6/INX have shown that a mutant INX-6 carrying an 18-amino-acid deletion in its amino-terminal intracellular domain forms constitutively closed gap junction channels (*Burendei et al., 2020*; *Oshima et al., 2016a*; *Oshima et al., 2016b*). In an attempt to generate a constitutively closed UNC-9 gap junction channel, we first examined the subcellular localization of UNC-9(ΔN18), which lacks the first 18 amino acids in its intracellular domain. Similar to wild type UNC-9::GFP, UNC-9(ΔN18)::GFP expressed in the DD neurons was localized at the anterior tip of DD6 axon, the putative gap junction site between DD5 and DD6 (*Figure 5A and B*). This result indicates that the 18 amino acid deletion does not alter either the synthesis or subcellular localization of the UNC-9(ΔN18) protein. We then tested if the putative channel-defective *unc-9(ΔN18)* can rescue the presynaptic tiling defect by expressing it under the *rgef-1* pan-neuronal promoter or under the *flp-13* DD-neuron-specific promoter. Surprisingly, both pan-neuronal and DD neuron-specific expression of *unc-9(Δ18N)* rescued the presynaptic tiling defect of *egl-20(n585); unc-9(e101)* mutants (*Figure 5D*). Importantly, pan-neuronal expression of wild type *unc-9* but not *unc-9(ΔN18)* rescued the kinker phenotype of the *unc-9(e101)* mutant (*Figure 4—figure supplement 1*). As the locomotion defects of *unc-9* mutants are largely due to the defective gap junction channel activity (*Kawano et al., 2011*), the failure to rescue the locomotion defects of *unc-9* mutants by *unc-9(Δ18N)* strongly suggests that UNC-9(ΔN18) forms a defective, likely a constitutively closed, gap junction channel.

To directly assess the effect of the ΔN18 deletion on UNC-9 gap junction channel activity, we tested whether expression of UNC-9(ΔN18) can rescue the electrical coupling defect of *unc-9(e101)* mutant. In these experiments, we expressed either wild type *unc-9* or *unc-9(ΔN18)* in the muscle cells using the muscle-specific *myo-3* promoter (P*myo-3*) in the *unc-9(e101)* mutant background. Cytoplasmic GFP was co-expressed in the muscle cells of these worms to serve as a transformation marker. We then performed dual whole-cell voltage clamp recordings on two contiguous neighboring ventral muscle cells from two different rows of muscle cells within the same quadrant (*Figure 5E*). Junctional currents ($I_j$) were measured from one muscle cell held constant at –30 mV while a series of membrane voltage steps (–110 to +50 mV at 10 mV intervals) were applied to the other neighboring muscle cell of the pair from a holding voltage of –30 mV (*Figure 5E*). We plotted the $I_j$ and transjunctional voltage ($V_j$) relationships and quantified the junctional conductance ($G_j$) from the slope of the linear portion of the $I_j$ - $V_j$ curves. While $I_j$ was prominent in the wild type, *unc-9(e101)* mutant showed very little $I_j$ (*Figure 5G*), similar to the *unc-9(fc16)* null mutants analyzed in our earlier studies (*Liu et al., 2011*; *Liu et al., 2006*). As expected, wild type UNC-9 restored the junctional current in the *unc-9(e101)* mutants, while *unc-9(e101)* mutants expressing *unc-9(ΔN18)* completely lacked the intra-quadrant coupling similar to *unc-9(e101)* mutants (*Figure 5F–H*). These results confirm that UNC-9(ΔN18) is unable to form functional gap junction channels, while maintaining the ability to control tiled presynaptic patterning between DD5 and DD6.

The successful rescue of the presynaptic tiling defect by the *unc-9(ΔN18)* transgenes could be due to the nature of overexpression of the transgene from extra-chromosomal concatemer arrays. To exclude this possibility, we generated *unc-9(ΔN18)* mutant using CRISPR/Cas9 genome editing. Similar to *unc-9(e101)* mutant, *unc-9(syb3236 [unc-9(ΔN18)])* mutant animals exhibit a kinker phenotype (*Figure 4—figure supplement 1*), suggesting that UNC-9(ΔN18) does not form functional gap junction channels. Despite the kinker phenotype, *unc-9(syb3236 [unc-9(ΔN18)])* mutant did not exhibit a presynaptic tiling defect between DD5 and DD6 in the *egl-20(n585)* mutant background (*Figure 5C and D*). The *unc-9(ΔN18)* deletion did not cause a presynaptic tiling defect in the *egl-20(n585); unc-1(miz297)* null mutant background where the UNC-9 gap junction channel activity is absent (*Figure 5D*). This further confirmed that UNC-9 does not require its gap junction channel activity to control presynaptic tiling.

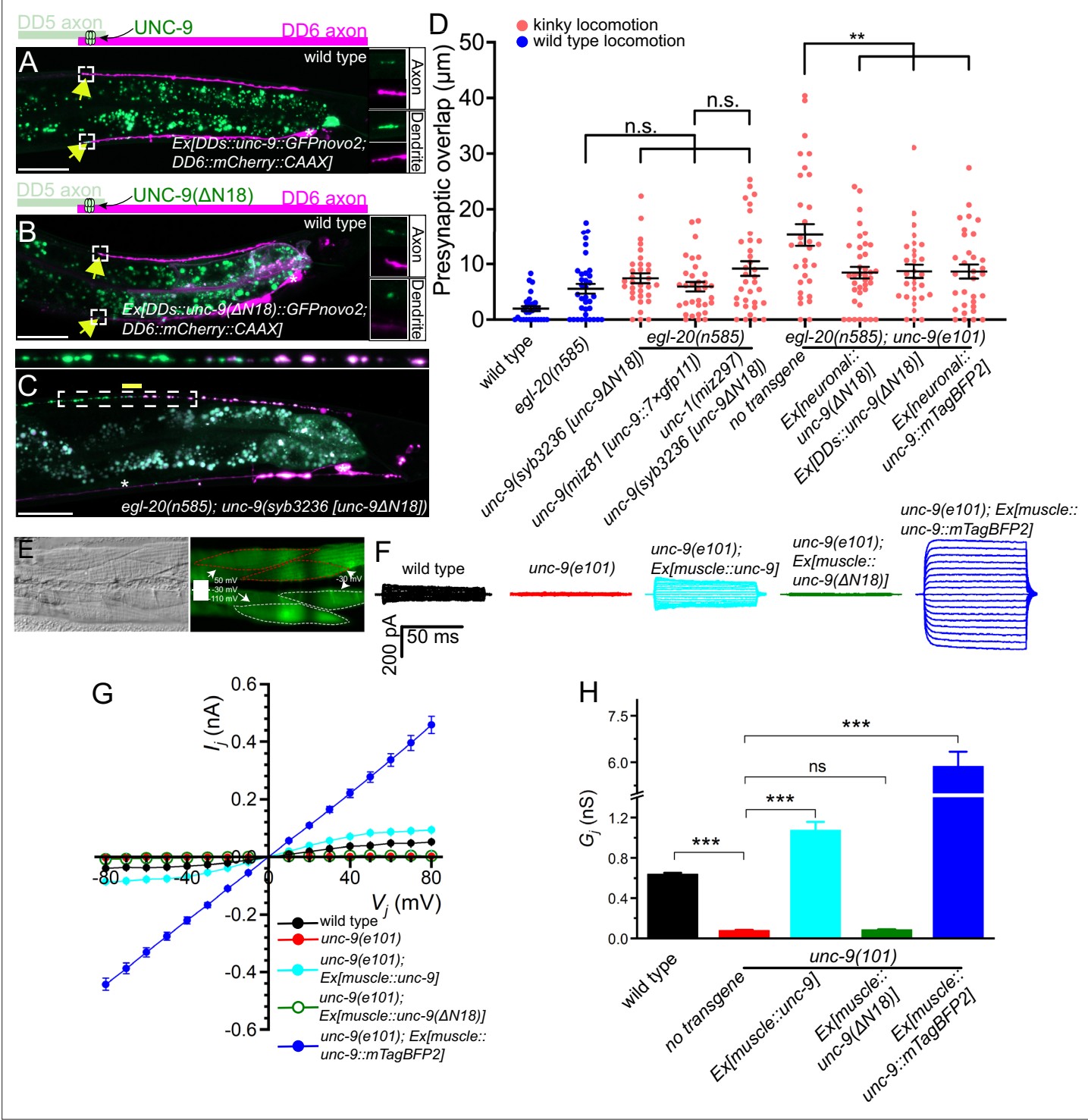

**Figure 5.** UNC-9 gap junction channel activity is dispensable for its function in presynaptic patterning. (**A–B**) Representative images of UNC-9::GFP (**A**) and UNC-9(ΔN18)::GFP (**B**) localization at the anterior tip of dorsal D-type 6 (DD6) axon and dendrite (indicated by yellow arrows) in wild type. The magnified UNC-9::GFP (**A**) and UNC-9(ΔN18)::GFP (**B**) and mCherry::CAAX signals in the anterior tip of DD6 axon and dendrite, indicated by the dashed box, are shown to the right of merged images. (**C**) Representative image of presynaptic tiling in the *egl-20(n585); unc-9(syb3236 [unc-9ΔN18])* double mutant. The magnified straightened image of the presynaptic tiling border, indicated by the dashed box, is shown above. Yellow line indicates region of presynaptic overlap between dorsal D-type 5 (DD5) and DD6. Asterisks: DD5 and DD6 cell bodies. Scale bar: 20 μm. (**D**) Quantification of presynaptic overlap between DD5 and DD6. See *Figure 5—source data 1*. Each dot represents a single animal. See source data for sample size and individual value. Black bars indicate mean ± SEM. n.s.: not significant; **p<0.01. (**E**) Representative image of adjacent body wall muscles expressing *unc-9* and

*Figure 5 continued on next page*

*Figure 5 continued*

GFPnovo2. (**F**) Sample traces of junctional currents ($I_j$) recorded from pairs of muscle cells like those surrounded by red or white dotted lines in panel E. (**G**) Graph of the junctional currents ($I_j$) and transjunctional voltage ($V_j$). See ***Figure 5—source data 2***. See source data for sample size and individual value. (**H**) Quantification of the junctional conductance ($G_j$). See ***Figure 5—source data 3***. See source data for sample size and individual value. Bars indicate mean ± SEM. n.s.: not significant; ***p<0.001.

The online version of this article includes the following source data for figure 5:

**Source data 1.** Quantification of overlap between DD5 and DD6 presynaptic domains.

**Source data 2.** Measurement of junctional currents ($I_j$) and transjunctional voltage ($V_j$).

**Source data 3.** Quantification of the junctional conductance ($G_j$).

## A putative, constitutively open form of UNC-9 can control presynaptic tiling

Previously, we showed that an UNC-9::GFP fusion protein may form constitutively open gap junction channels in muscles and that such gap junctions do not require UNC-1 to function (***Chen et al., 2007***). The dispensability of UNC-1 to UNC-9::GFP gap junctions has also been shown with *C. elegans* neurons (***Jang et al., 2017***). To further substantiate the potential channel-independent function of UNC-9 in controlling presynaptic tiling, we examined whether *unc-9::mTagBFP2* can rescue the presynaptic tiling defect. We chose UNC-9::mTagBFP2 instead of UNC-9::GFP, as the GFP signal interferes with our presynaptic marker (GFP::RAB-3). We first tested whether UNC-9::mTagBFP2 forms an open gap junction channel similar to UNC-9::GFP by testing whether it can rescue the electrical coupling defects of *unc-9(e101)* mutant. The muscle expression of *unc-9::mTagBFP2* in *unc-9(e101)* mutant resulted in a much larger $I_j$ and $G_j$ than those obtained with wild type *unc-9* (***Figure 5F–H***), which is reminiscent of our previous observation with *unc-9::GFP* (***Chen et al., 2007***). We then expressed *unc-9::mTagBFP2* pan-neuronally using the *rgef-1* promoter, and tested whether it has the ability to control presynaptic tiling patterning. Similar to *unc-9(ΔN18)*, pan-neuronal expression of *unc-9::mTagBFP2* rescued the presynaptic patterning defect of *egl-20(n585); unc-9(e101)* mutant (***Figure 5D***). Importantly, pan-neuronal expression of *unc-9::mTagBFP2* did not rescue the kinker phenotype of *unc-9(e101)* mutant (***Figure 4—figure supplement 1***), which is probably because hyperactive gap junction channel activity formed by UNC-9::mTagBFP2 impaired rather than restored the function of the locomotor neural circuit. We further confirmed this result by examining the presynaptic tiling pattern in the *egl-20(n585); unc-9(miz81 [unc-9::7×gfp11])* mutant. UNC-9::7×GFP11, which by itself is not fluorescent, has seven tandem repeats of GFP11 at the carboxy-terminus of UNC-9, and hence, is expected to have a similar effect on gap junction function as UNC-9::GFP or UNC-9::mTagBFP2. While UNC-9::7×GFP localizes to the putative gap junction sites between DD5 and DD6 axons, *unc-9(miz81 [unc-9::7×gfp11])* mutants exhibit a kinker phenotype (***Figure 4—figure supplement 1***), suggesting that they form defective, likely an open form of gap junctions. Despite the kinker phenotype, *egl-20(n585); unc-9(miz81 [unc-9::7×gfp11])* mutant does not exhibit the presynaptic tiling defect (***Figure 5D***). These results suggest that an increased activity of UNC-9 gap junctions does not compromise the physiological role of UNC-9 in controlling the presynaptic tiling pattern. Taken together, our results indicate that UNC-9 controls presynaptic tiling through a gap junction channel-independent function.

## Discussion

Here we established a novel system to study neuronal and synaptic tiling in the DD type GABAergic motor neurons in *C. elegans*. We found that EGL-20/Wnt negatively regulates the length of the posterior axon and dendrite of DD5. We showed that presynaptic tiling requires UNC-9/INX, while the position of postsynaptic spines appears to depend on the length of the posterior dendrite. UNC-9 is localized at the presynaptic tiling border between DD5 and DD6 where it forms gap junction channels. Strikingly, the gap junction channel activity of UNC-9 is dispensable for its function in establishing presynaptic tiling.

### Redundant actions of Wnt and INX in establishing DD presynaptic tiling

In wild type animals, the presynaptic domain of each DD neuron does not overlap with those from neighboring DDs, because of their tiled axonal patterning. However, our observation revealed that

the axonal tiling is dispensable for the presynaptic tiling, which is controlled by UNC-9. Therefore, the presynaptic tiling of the DD neurons is governed by two redundant pathways: Wnt-dependent axonal tiling and INX-dependent presynaptic tiling. This observation is very unique compared to other systems such as the L1 lamina neuron in *Drosophila* where axonal tiling defects cause disrupted synaptic connections (*Millard et al., 2007*). DD neurons are critical for the coordinated contractions and relaxations between the dorsal and ventral body wall muscles during sinusoidal locomotion of the worms (*Kawano et al., 2011*). As the synapse is the functional unit for this functionality of the DD neurons, the redundant mechanisms to tile synapses in DD neurons by Wnt and INX may ensure the proper functions of the DD neurons. This may be because the DD neurons are the only GABAergic-class of motor neurons in the dorsal nerve cord and thus require more robust control of their synaptic patterning.

The degree of presynaptic tiling defect of *egl-20(n585); unc-9(e101)* mutant is smaller than the degree of axonal tiling defect, suggesting the presence of additional factors that regulate presynaptic patterning between DD5 and DD6 along with *unc-9*. Previously, we showed that spatial expression pattern of two Wnt proteins (LIN-44 and EGL-20) defines topographic presynaptic patterning of DA8 and DA9 in the absence of Plexin-dependent presynaptic tiling mechanism (*Mizumoto and Shen, 2013b*). It is therefore possible that LIN-44/Wnt contributes toward the positioning of DD5 synapses in the *egl-20; unc-9* mutant. We could not test this hypothesis as the expression pattern of the DD5/DD6 presynaptic tiling marker is disrupted in the *lin-44* mutant.

## Mechanisms of INX-dependent synapse patterning and potential links to synaptopathies

How does UNC-9 control presynaptic tiling between DD5 and DD6? UNC-9 gap junctions are localized at the presynaptic tiling border in both wild type and *egl-20* mutant. The specific subcellular localization of UNC-9 gap junctions at the presynaptic tiling border suggests that UNC-9 acts as a positional cue by forming a highly localized gap junction plaque that locally restricts synapse formation. Indeed, loss of *unc-9* results in the ectopic synapse formation in the distal posterior axonal region of DD5. Consistent with our observations, recent work in mouse cortical neurons showed that PANX1 inhibits synapse formation (*Sanchez-Arias et al., 2020*; *Sanchez-Arias et al., 2019*). It is therefore possible that gap junction proteins have conserved functions as negative regulators for synapse formation. Previously, UNC-9 was shown to localize at the nonjunctional perisynaptic region in the DD neurons to regulate active zone differentiation, possibly as hemichannels (*Yeh et al., 2009*). *unc-9*'s function in promoting presynaptic assembly contrasts with our findings that *unc-9* restricts synapse formation. It is possible that distinct localization of UNC-9 may inhibit or promote synapse formation through distinct downstream molecular pathways. While we do not know how UNC-9 locally restricts synapse formation, it is possible that it acts through actin cytoskeleton and its regulators, as proper actin cytoskeleton formation is essential for synapse formation (*Aiken and Holzbaur, 2021*; *Hendi et al., 2019*). Cx proteins have been shown to interact with various actin cytoskeletal regulators including Rap1, IQGAP, ZO-1, Claudin, and Drebrin (*Falk et al., 2014*; *Lasseigne et al., 2021*; *Olk et al., 2009*). Indeed, we have previously shown that RAP-2 small GTPase and its effector kinase MIG-15/TNIK is crucial for the presynaptic tiling of the DA8 and DA9 neurons (*Chen et al., 2018*). However, we did not observe DD5/DD6 presynaptic tiling defects in the mutants of *rap-1/Rap1, rap-2/Rap2, pes-7/IQGAP, zoo-1/ZO-1, vab-9/Caludin, dbn-1/Drebrin,* and *mig-15/TNIK* in the *egl-20(n585)* background (*Figure 2—figure supplement 1*). Therefore, the DD5 and DD6 neurons seem to control presynaptic tiling through an uncharacterized mechanism.

Mutations in gap junction proteins are often associated with various synaptopathies caused by abnormal synapse number and position (*Lapato and Tiwari-Woodruff, 2018*). The present work that revealed the novel function of a gap junction protein in presynaptic patterning will help us better understand how mutations in gap junction proteins cause synaptopathies. Furthermore, candidate and forward screenings are essential to uncover the molecular mechanisms that underlie INX-dependent synapse patterning.

## Limitation of the present work

This work showed that the presynaptic tiling of DD neurons is controlled by Wnt-dependent axonal tiling and UNC-9-dependent presynaptic tiling. The redundant mechanisms to set up tiled presynaptic

arrangement argue that it is important for the function of these neurons in locomotion. However, we could not directly test the effect of disrupted presynaptic tiling of the DD GABAergic motor neurons on locomotion, as all mutants with disrupted presynaptic tiling exhibited a kinker phenotype due to defective UNC-9-channel activity. Creating *unc-9* mutants which specifically disrupt UNC-9's function in presynaptic tiling without affecting its channel activity will help us to uncover the functional importance of the presynaptic tiling of the DD neurons.

DD neurons undergo synaptic remodeling at the end of the first larval stage L1, when the dorsal dendrites switch their fates to axons to form presynaptic connections with the dorsal body wall muscles (*White et al., 1978*). We showed that at mid L2 stage, *egl-20(n585); unc-9(e101)* double mutants exhibit presynaptic tiling defects (*Figure 2—figure supplement 3*). This suggests that *egl-20* and *unc-9* are required for the establishment of axonal and presynaptic tiling, respectively. However, due to the small size of the L1 animals, we could not observe the presynaptic tiling patterning during DD remodeling to observe how presynaptic tiling is established.

We did not identify UNC-9's downstream effectors (see above), nor the upstream components that are required for UNC-9 localization at the presynaptic tiling border. There are several genes that are necessary for the proper clustering of gap junction channels. However, UNC-9 localization was unaffected in the mutants of *zoo-1/ZO-1* and *nlr-1/CASPR*. Recent work showed that cAMP-dependent axonal transport is required for the proper localization of UNC-9 in the VA (ventral A-class) cholinergic motor neurons in *C. elegans* (*Palumbos et al., 2021*). We observed a slight reduction in UNC-9 signal intensity at the presynaptic tiling border in the *unc-104/Kif1A* mutants, suggesting that UNC-9 transport is partly dependent on *unc-104*. The stereotypical localization of UNC-9 at the presynaptic border between DD neurons provides an ideal platform to carry out genetic screens to decipher the molecular mechanisms that underlie gap junction localization.

## Materials and methods

### Strains

Bristol N2 strain was used as wild type reference. All strains were cultured in the nematode growth medium with OP50 as described previously (*Brenner, 1974*) at 25°C. The following alleles were used in this study: *dbn-1(ok925)*, *egl-20(n585)*, *inx-1(miz280)*, *inx-2(miz282)*, *inx-10(miz284)*, *inx-14(ok3267)*, *mig-15(rh148)*, *nlg-1(yv15)*, *nlr-1(miz202)*, *pes-7(gk123)*, *plx-1(nc36)*, *rap-1(miz203)*, *rap-2(gk11)*, *unc-1(e719)*, *unc-1(miz297)*, *unc-7(e5)*, *unc-9(e101)*, *unc-9(tm5479)*, *unc-9(miz81 [unc-9::7×gfp11])*, *unc-9(syb3236 [unc-9ΔN18])*, *unc-24(miz225)*, *unc-24(miz226)*, *unc-104(e1265)*, *nlr-1(gk366849)*, *vab-9(ju6)*, *zoo-1(tm4133)*. Genotyping primers are listed in the supplemental material.

### Transgenes

The transgenic lines with extrachromosomal arrays were generated using the standard microinjection method (*Fire, 1986*; *Mello et al., 1991*). The integration of the extrachromosomal arrays into the chromosomes was conducted by standard UV irradiation method (*Evans, 2006*).

The following transgenes were used in this study: *mizIs3* (P*unc-4c::zf1::GFPnovo2::rab-3*, P*mig-13::zif-1*, P*mig-13::mCherry::rab-3*); *wyIs292* (P*unc-47::unc-10::tdTomato*, P*unc-129(dorsal muscle)::nlg-1::yfp*); *wyIs442* (P*flp-13::2×GFP::rab-3*, P*plx-2::2×mCherry::rab-3*); *wyIs486* (P*flp-13::2×GFP*, P*plx-2::2×mCherry*); *juIs463* (P*flp-13::GFP1-10*, P*ttx-3::RFP*); *ufIs126* (P*flp-13::acr-12::GFP*); *mizEx69*, *mizEx407* (P*rgef-1::unc-9*, P*odr-1::GFP*); *mizEx72*, *mizEx420* (P*rgef-1::unc-9(ΔN18)*, P*odr-1::GFP*); *mizEx370*, *mizEx371* (P*flp-13::BFP::rab-3*, P*odr-1::GFP*); *mizEx416* (P*flp-13::unc-9*, P*odr-1::GFP*); *mizEx429*, *mizEx433* (P*flp-13::unc-9(ΔN18)*, P*odr-1::GFP*); *mizEx430* (P*flp-13::mCherry::rab-3*, P*odr-1::GFP*); *mizEx510* (P*myo-3::unc-9*, P*odr-1::GFP*), *mizEx496*, *mizEx497* (P*myo-3::unc-9*, P*myo-3::GFPnovo2*, P*odr-1::GFP*); *mizEx499*, *mizEx500* (P*myo-3::unc-9(ΔN18)*, P*myo-3::GFPnovo2*, P*odr-1::GFP*); *mizEx498*, *mizEx499* (P*myo-3::unc-9(ΔN18)*, P*myo-3::GFPnovo2*, P*odr-1::GFP*); *mizEx500*, *mizEx501* (P*myo-3::unc-9::mTagBFP2*, P*myo-3::GFPnovo2*, P*odr-1::GFP*); *mizEx515* (P*egl-20::egl-20*, P*odr-1::GFP*); *mizEx543*, *mizEx544* (P*flp-13::GFP::unc-9*, P*plx-2::mCherry::CAAX*, P*odr-1::GFP*); *mizEx546*, *mizEx547* (P*hsp::unc-9*, P*odr-1::GFP*); *mizEx548*, *mizEx549* (P*flp-13::mCherry::CAAX*, P*odr-1::GFP*); *mizEx550*, *mizEx551* (P*flp-13::mCherry::rab-3*, *mizEx548*, *mizEx549* (P*flp-13::2×mTagBFP2*, P*odr-1::GFP*); P*odr-1::GFP*); *mizEx558*, *mizEx559* (P*itr-1::GFP1-10*, P*itr-1::mCherry::CAAX*, P*odr-1::GFP*).

## Key strains

UJ1044 *egl-20(n585); wyIs442*
UJ1215 *egl-20(n585); unc-9(e101); wyIs442*
UJ1543 *egl-20(n585); unc-9(syb3236 [unc-9ΔN18]); wyIs442*
UJ1261 *unc-9(miz81 [unc-9::7×gfp11]); juIs463*

## Plasmid construction

*C. elegans* expression clones were made in a derivative of pPD49.26 (A. Fire), the pSM vector (a kind gift from S. McCarroll and C. I. Bargmann). *unc-9* cDNAs were amplified with Phusion DNA polymerase (NEB) from N2 cDNA library synthesized with superscript III first-strand synthesis system (Thermo Fisher Scientific). The amplified cDNAs were cloned into the AscI and KpnI sites of pSM vector using Gibson assembly method (*Gibson, 2011*). Plasmid design was conducted on A plasmid Editor (*Davis and Jorgensen, 2022*).

## Bashed *plx-2* promoter construction for DD6-specific expression

A 4.5-Kb PCR fragment of the 5' promoter region of *plx-2* was cloned upstream of GFP between *Hinc*II and *Msc*I in pPD95.75, to generate pRI20. A sub-fragment of the *plx-2* promoter that drives selective expression in the DD6 neuron was generated by deleting a 3587 bp internal region in the plx-2 promoter of pRI20, by digesting with *Hpa*I and *Bam*HI, end-filling the *Bam*HI 5' overhang with Klenow, and blunt-end ligation to generate pRI50. The resulting *plx-2* promoter was re-cloned into the *Sph*I and *Asc*I sites of the pSM vector.

## CRISPR/Cas9 genome editing

The *loxP::myo-2::NeoR* dual-selection cassette vector (*Au et al., 2019*; *Norris et al., 2015*) was used to construct the repair template plasmids for *unc-9(miz81 [unc-9::7×gfp11])*. The *7×gfp11* sequence was cloned into the *Sac*II site of the *loxP::myo-2::NeoR* plasmid using Gibson assembly (*Gibson, 2011*) The 5' and 3' homology arms of the *unc-9* repair template were amplified from N2 genomic DNA using Phusion DNA polymerase and cloned into the *Sac*II and *Not*I restriction sites, respectively, by Gibson assembly.

pTK73 plasmid was used as a backbone vector for gRNA expression plasmid construction for *unc-9(miz81 [unc-9::7×gfp11])* strain (*Obinata et al., 2018*). The following guide RNA was used *unc-9#1*: AATTAAACCCCATTTCAGGA.

To generate *unc-9(miz81 [unc-9::7×gfp11])*, the *unc-9* repair template plasmid, *unc-9* sgRNA plasmid, and Cas9 plasmid (*Friedland et al., 2013*) were co-injected into young adults. The screening of the genome-edited animals was conducted as previously described (*Au et al., 2019*; *Norris et al., 2015*). F1 progenies were treated with Geneticin (G418) (Sigma-Aldrich) (50 mg/mL) for NeoR selection. Animals with uniform pharyngeal expression of P*myo-2::GFP* were selected as genome-edited candidates. The selection cassette was excised out by Cre recombinase (pDD104, Addgene #47551). Progenies without pharyngeal P*myo-2::GFP* expression were isolated. Successful genome-edited candidate animals were confirmed via PCR and Sanger sequencing.

To generate *inx-1(miz280), inx-2(miz282), inx-10(miz284), nlr-1miz202 rap-1(miz203), unc-1(miz297)*, and *unc-24(miz225)* & (*miz226*) mutants, we used a direct injection of gRNA/Cas9 complex with the homology-dependent repair (HDR) donor oligos (*Dokshin et al., 2018*). *unc-24(miz225)* and (*miz226*) alleles have an identical mutation.

The HDR donor template oligos include mutations that introduce premature stop codon and frameshift, along with either *Eco*RI or *Bam*HI sites for genotyping purpose. Successful genome-edited candidate animals were confirmed by PCR and Sanger sequencing.

The mutations introduced in these mutants, and the sequences of gRNA and HDR donor oligos are as follows:

*inx-1(miz280)*: Gly222Stop
*inx-2(miz282)*: Gly138Stop
*inx-10(miz284)*: Phe138Stop
*nlr-1(miz202)*: Asn325Stop

*rap-1(miz203)*: Ser17Stop
*unc-1(miz287)*: Isoform A-Ser128Stop;
Isoform B-Ser132Stop
*unc-24(miz225)* & (*miz226*): Ala281Stop
*inx-1*
gRNA: GGTACTGACAACCTTTTTTA
HDR donor oligos:
Forward:CAAGTGTTTATGTTAAACAGTTTCCTTGGTACTGACAACCTTTTTTACtgagaattcTTC
ACATTTTGAGAGATTTGTTGAATGGTCGTGAATG
Reverse:CATTCACGACCATTCAACAAATCTCTCAAAATGTGAAgaattctcaGTAAAAAAGGTTGT
CAGTACCAAGGAAACTGTTTAACATAAACACTTG
*inx-2*
gRNA: AGAACCACAATTAACCTCAA
HDR donor oligos:
Forward:CATCTCTGGAATTTGTTTCACAAAAGAACCACAATTAACCTCAAAtgagaattcCTTTAA
GATTTTTTGAAGGAGCTTTGAAAAAATTGGAG
Reverse:CTCCAATTTTTTCAAAGCTCCTTCAAAAAATCTTAAAGgaattctcaTTTGAGGTTAATT
GTGGTTCTTTTGTGAAACAAATTCCAGAGATG
*inx-10*
gRNA: ACGGTAGATTGAAAAGCCAA
HDR donor oligos:
Forward:CATCGACGCCTGCAAAAGAAACAAATTCGTCCGCATCGTTgagaattcTTGGCTTT
TCAATCTACCGTACTCTGCATTCTTTGTGACTGCAATG
Reverse:CATTGCAGTCACAAAGAATGCAGAGTACGGTAGATTGAAAAGCCAAgaattctcAA
CGATGCGGACGAATTTGTTTCTTTTGCAGGCGTCGATG
*nlr-1*
gRNA: GTTACAAGTGGACCATTGAT
HDR donor oligos:
Forward:CATTCACTTTCCTTCCGTTTTCATTCGAGTTCCGGATAGTTgaattcTAATCAATGGTCC
ACTTGTAACACTTTTTGATGCTGAAAATGGAAC
Reverse:GTTCCATTTTCAGCATCAAAAAGTGTTACAAGTGGACCATTGATTAgaattcAACTATCC
GGAACTCGAATGAAAACGGAAGGAAAGTGAATG
*rap-1*
gRNA: AAGATTGTTGTGCTCGGATC
HDR donor oligos:
Forward:caaaaaaagaATGCGGGAGTATAAGATTGTTGTGCTCGGATCTGaggatccgGAGGAGTA
GGAAAATCCGCACTGgtatagatgagagtgaccgg
Reverse:ccggtcactctcatctatacCAGTGCGGATTTTCCTACTCCTCcggatcctCAGATCCGAGCAC
AACAATCTTATACTCCCGCATtcttttttg
*unc-1*
gRNA: ACCCGATTGCTTCCGTCAAC
HDR donor oligos:
Forward:GTATCCGTCGACGCCGTTGTCTACTTCCGTACCTgagaattcACCCGATTGCTTCC
GTCAACAATGTTGACGATGCCATTTACTCCACCAAACTGC
Reverse:GCAGTTTGGTGGAGTAAATGGCATCGTCAACATTGTTGACGGAAGCAATCGGGTga
attctcAGGTACGGAAGTAGACAACGGCGTCGACGGATAC
*unc-24*
gRNA: CGTTGAGCAGCGTTGCGAAA
HDR donor oligos:
Forward:GGAAGGAGAGAACATGGGGATGTCTGCGTTGAGCAGCGTTGCGAAAgaattcTGAT
GCTGGTCAACAGTTGTGGCAAGTTATTGGACCAGTATTCG
Reverse:CGAATACTGGTCCAATAACTTGCCACAACTGTTGACCAGCATCAgaattcTTTCGC
AACGCTGCTCAACGCAGACATCCCCATGTTCTCTCCTTCC

## Confocal and stereo microscopy

Images of fluorescently tagged fusion proteins used here (BFP, GFP and mCherry) were captured in live *C. elegans* using a Zeiss LSM800 Airyscan confocal microscope (Carl Zeiss, Germany) with oil immersion lens 40× and 63× magnification (Carl Zeiss, Germany). Worms were immobilized on 5% agarose pads using a 3:1 mixture of 0.225 M 2,3-butanedione monoxime (Sigma-Aldrich) and 7.5 mM levamisole (Sigma-Aldrich). Images were analyzed using Zen software (Carl Zeiss). Images were straightened with ImageJ (NIH, USA). A range of 18–24 Z-stack images were taken for each animal to encompass the cell bodies, axons, and synapses of the DD5 and DD6 neurons. L4.4–L4.6 larval stage animals, judged by the stereotyped shape of the developing vulva (*Mok et al., 2015*), were used for quantification.

Stereoscope images were taken on Zeiss Stemi 305 with Zeiss Labscope.

## Fluorescent signal intensity quantification

For quantifying fluorescent signal intensity of UNC-9::7×GFP, 21 Z-stacks are compiled using ImageJ (Image → Stacks → Z-project → Sum Slices). The signal intensity of 7.33 µm of region of interest (ROI) at the anterior tip of the DD6 axon is measured using ImageJ (Analyze → Plot Graph). The signal from the region adjacent to the ROI is used as background signal which was subtracted from the signal intensity of UNC-9::7×GFP.

## Statistics

Generated data were analyzed and processed by Prism9 (GraphPad Software, USA). We applied the one-way ANOVA method with post hoc Tukey's multiple comparison test (or Fisher's post hoc test for *Figure 5G*) for comparison among three or more parallel groups with multiple plotting points. T-test was used for comparison between two binary points. Data were plotted with error bars representing standard errors of mean (*Jörgensen et al., 2022*). *, **, and *** represent p-value <0.05, <0.01, <0.001, respectively.

## Electrophysiology

Adult (day 1) hermaphrodite animals were immobilized and dissected as described previously (*Liu et al., 2011*; *Liu et al., 2006*). Briefly, an animal was immobilized on a glass coverslip by applying Vetbond Tissue Adhesive (3 M Company, MN, USA). Application of the adhesive was generally restricted to the dorsal middle portion of the animal, allowing the head and tail to sway during the experiment. A longitudinal incision was made in the dorsolateral region. After clearing the viscera, the cuticle flap was folded back and glued to the coverslip, exposing the ventral nerve cord and the two adjacent muscle quadrants. A Nikon FN-1 microscope equipped with a 40× water-immersion objective and 15× eyepieces was used for viewing the preparation. Borosilicate glass pipettes with a tip resistance of 3–5 MΩ were used as electrodes for voltage-clamp recordings in the classical whole-cell patch clamp configuration with a Multiclamp 700B amplifier (Molecular Devices, Sunnyvale, CA) and the Clampex software (version 11, Molecular Devices). To record $I_j$, the membrane potential ($V_m$) of both cells was held at –30 mV, from which a series of voltage steps (–110 mV to +50 mV at 10 mV intervals and 100ms duration) were applied to one cell (Cell 1), whereas the other cell (Cell 2) was held constant to record $I_j$. $V_j$ was defined as $V_m$ of Cell 2 minus $V_m$ of Cell 1. Series resistance was compensated to approximately 80% in the voltage-clamp experiments. Data was sampled at a rate of 10 kHz after filtering at 2 kHz. The bath solution and pipette solution used were identical to those described earlier (*Liu et al., 2011*; *Liu et al., 2006*).

## Genotyping primers

*dbn-1(ok925)*
Forward: 5' CTGCGCTTATTATCGTCCTG 3'
Wild type reverse: 5' CGATCAAGTCGATAGCGGATG 3'
Mutant reverse: 5' GCACGGCCGCGGCAATTCACAC 3'
*egl-20(n585)* – wild type PCR product will be digested by *HpyCH*4V
Forward: 5' CTCTTAAAAACTTACCTCTCAAATTTGAACTTATTCTTGC 3'
Reverse: 5' CCTCATTACCATTCAACTGATAG 3'

*inx-1(miz280)* – mutant PCR product will be digested by *Eco*RI
Forward: 5' CAATGTGCAATCATTGACTCAAATGGC 3'
Reverse: 5' CATCATACAGTCCAATGCGTGCTAATG 3'
*inx-2(miz282)* – mutant PCR product will be digested by *Eco*RI
Forward: 5' GTATGAGCAACTAGTAGGGATGCTGCG 3'
Reverse: 5' CCTAACTTGAGCAAGAAGTAATTGATGGTAGCTTG 3'
*inx-10(miz284)* – mutant PCR product will be digested by *Eco*RI
Forward: 5' GAGATTGTCAAGTTATCTTCGGATCCC 3'
Reverse: 5' CCAGCACACATTGTATAGTATGTTC 3'
*nlr-1(miz202)* – mutant PCR product will be digested by *Eco*RI
Forward: 5' GTTGTAAGGTCAACTATTAGAGCAAGG 3'
Reverse: 5' GTTCGAGTTGATGGAAACGTGACAAG 3'
*nlr-1(gk366849)* – wild type PCR product will be digested by *Mfe*I
Forward: 5' GTTTGCTCTCTTCATCAATCACTACATCC 3'
Reverse: 5' CGCCATAAAACGATATATTATGTGTAG 3'
*pes-7(gk123)*
Forward: 5' GTGTAGGTGTGAGGAAGTCC 3'
Wild type reverse: 5' CGGCGTCGCATCATTCTGCCG 3'
Mutant reverse: 5' CCGCCAATTAGATTTGTTCG 3'
*plx-1(nc36)*
Forward: 5' CTTCGAGAGCCCCCTCATTCTTAATG 3'
Reverse 5' CCGGCACACGTTAAACTAGTGCTACCG 3'
*rap-1(miz203)* – mutant PCR product will be digested by *Bam*HI
Forward: 5' GTGTCATCTGGTCTGTACTTGGTGC 3'
Reverse: 5' GGAGTACTGTAACTAAAAACTCTCGTC 3'
*rap-2(gk11)*
Forward: 5' TCTCATCTCCATCGTCGTTCCTGC 3'
Wild type reverse: 5' TCCATTCACTGAATGTTCCGC 3'
Mutant reverse: 5' GAGGGAGTTCAAAGTGGTCGTTC 3'
*unc-1(miz297)* – mutant PCR product will be digested by *Eco*RI
Forward: 5' CTGACCTGGGCTCACTTTTAAGTC 3'
Reverse: 5' GCCCCAAATATGAGCATTGAGTGATCAG 3'
*unc-24(miz225)* & (*miz226*) – mutant PCR product will be digested by *Eco*RI
Forward: 5' CCACAGA TCGTGGTCTCGTGGAAC 3'
Reverse: 5' CTGACATTCGCTCCACCAAGTGTTTTAGC 3'
*zoo-1(tm4133)*
Mutant forward: 5' CAGGTCGGCGGAAGTGTCGGAGTACGTG 3'
Wild type forward: 5' CCGAATCAAGCGACCGCCGAGCAAATTGC 3'
Reverse: 5' GTGCCAGCTGAAGACGTTCAACAGACTCG 3'

## Acknowledgements

We would like to thank Cornelia I Bargmann, Catharine H Rankin, and Michael M Francis for sharing plasmids and strains, Shinsuke Niwa for sharing pTK73 gRNA plasmid, Atsunori Oshima for conducting preliminary experiments, Donald G Moerman, Leigh Anne Swayne and the Mizumoto lab members for discussions. *unc-9(syb3236 [unc-9ΔN18])* was generated by SunyBiotech, China. Some strains used in this study were obtained from the *Caenorhabditis* Genetics Center, CGC, funded by National Institute of Health (NIH) - Office of Research Infrastructure Programs (P40 OD010440), *C. elegans* gene knockout consortium, and the National Bioresource Project, Japan. This work is supported by HFSP CDA-00004–2014 (K.M.), CIHR AWD-017638 (K.M.), CIHR PJT- 180563 (K.M.), NIH R01NS109388 (Z.W.), and NIH R01MH085927 (Z.W.). K.M. is a recipient of Canada Research Chair and Michael Smith Foundation for Health Research Scholar. A.H. is a recipient of NSERC CGS-D and the UBC 4-year fellowships.

## Additional information

### Funding

| Funder | Grant reference number | Author |
|---|---|---|
| Canadian Institutes of Health Research | AWD-017638 | Kota Mizumoto |
| Canadian Institutes of Health Research | PJT- 180563 | Kota Mizumoto |
| National Institutes of Health | R01NS109388 | Zhao-Wen Wang |
| National Institutes of Health | R01MH085927 | Zhao-Wen Wang |
| Human Frontier Science Program | CDA-00004-2014 | Kota Mizumoto |
| Canada Research Chairs | 950-232435 | Kota Mizumoto |
| Michael Smith Foundation for Health Research | SCH-2017-2001 | Kota Mizumoto |

The funders had no role in study design, data collection and interpretation, or the decision to submit the work for publication.

### Author contributions

Ardalan Hendi, Conceptualization, Data curation, Formal analysis, Validation, Investigation, Visualization, Methodology, Writing – original draft, Writing – review and editing; Long-Gang Niu, Data curation, Formal analysis, Investigation, Visualization, Methodology, Writing – original draft, Writing – review and editing; Andrew William Snow, Data curation, Formal analysis, Investigation, Methodology, Writing – review and editing; Richard Ikegami, Resources, Methodology, Writing – original draft; Zhao-Wen Wang, Funding acquisition, Investigation, Methodology, Writing – original draft, Writing – review and editing; Kota Mizumoto, Conceptualization, Supervision, Funding acquisition, Validation, Investigation, Methodology, Writing – original draft, Writing – review and editing

### Author ORCIDs

Ardalan Hendi ⓘ http://orcid.org/0000-0001-6153-4266
Long-Gang Niu ⓘ http://orcid.org/0000-0001-7209-7436
Zhao-Wen Wang ⓘ http://orcid.org/0000-0003-3574-8556
Kota Mizumoto ⓘ http://orcid.org/0000-0001-8091-4483

### Decision letter and Author response

Decision letter https://doi.org/10.7554/eLife.80555.sa1
Author response https://doi.org/10.7554/eLife.80555.sa2

## Additional files

### Supplementary files

• Transparent reporting form

### Data availability

We include all source data for each quantification in the figures.

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
