## [Editor Report]

This work provides a highly valued addition to our understanding of innexin gene function in the nervous system. The authors describe here potential functions in synapse tiling. The paper should be of interest to researchers with an interest in molecular mechanisms governing nervous system development.

---

## [Decision Letter]

**Decision letter after peer review:**

Thank you for submitting your article "Channel-independent function of UNC-9/INX in spatial arrangement of GABAergic synapses in *C. elegans*" for consideration by *eLife*. Your article has been reviewed by 3 peer reviewers, and the evaluation has been overseen by a Reviewing Editor and Piali Sengupta as the Senior Editor. The following individual involved in review of your submission has agreed to reveal their identity: Michael M Francis (Reviewer #3).

The reviewers have discussed their reviews with one another.

As you will see in the reviews below, all three reviewers thought that this paper has very good potential, but that a number of revisions are required to make the paper become potentially acceptable. Each one of the 3 reviewers has non-overlapping sets of suggestions and we all felt that in aggregate these revisions are all doable (in regard to one point: expression patterns are fine to infer from an scRNA atlas, no new reagents need to be generated).

*Reviewer #1 (Recommendations for the authors):*

This reviewer suggests using heat shock promoter to express UNC-9::GFP to check whether late expression of UNC-9 could revert the presynaptic pattering defect observed in egl-20; unc-9 double mutant animals.

*Reviewer #2 (Recommendations for the authors):*

A few questions need to be addressed for current version of the manuscript.

1) Confirm UNC-9 localization using anti-UNC-9 antibody. As evidenced in this study UNC-9::GFP is a gain-of-function version of UNC-9, and it may or may not represent UNC-9 localization in vivo.

2) Test whether unc-9 is also required for DD5/DD6 synapse tilling in other genetic backgrounds, in which DD5 and DD6 have overlapping axons. one concern is whether unc-9 is only required for synapse tilling when egl-20 is mutated.

3) DD5/DD6 don't have overlapping axons in wild type animals, and therefore all experiments for DD5/DD6 synapse tilling can only be carried out in mutants with abnormally overlapped DD5/DD6 axons. One concern is how "synapse tilling" in these abnormally developed axons represents synapse tilling during normal development. I understand it is challenging to address this in DD neurons, but it should be testable in other neurons who have overlapping axons and tilled synapses, for example DA neurons.

*Reviewer #3 (Recommendations for the authors):*

My major comment relates to the potential advance offered by the studies. The impact of the findings would be enhanced if the authors were able to define mechanisms for UNC-9 localization and/or function in tiling. Nonetheless, the findings of the study as presented are provocative and suggest a potentially novel mechanism for synaptic tiling through regulated expression/localization of gap junction proteins. This will be of interest to the field and sets the stage for follow up studies of mechanism.

In several instances, the writing is unclear:

Line 148: "we observed minimal overlaps between the axons and the dendrites of DD5 and DD6" would be more clearly stated as "minimal overlap between either axons or dendrites of…"

Lines 168-9 are redundant with lines 173-4

Lines 206-7: "the position of DD5 postsynaptic dendritic spine is determined by the length of DD5 dendrite". This is unclear, ACR-12::GFP puncta appear to extend along the full length of the dendrite.

Lines 275-6: Delete "both".

---

## [Author Response]

Reviewer #1 (Recommendations for the authors):This reviewer suggests using heat shock promoter to express UNC-9::GFP to check whether late expression of UNC-9 could revert the presynaptic pattering defect observed in egl-20; unc-9 double mutant animals.

We have conducted this experiment using a heat-shock promoter.

Reviewer #2 (Recommendations for the authors):A few questions need to be addressed for current version of the manuscript.1) Confirm UNC-9 localization using anti-UNC-9 antibody. As evidenced in this study UNC-9::GFP is a gain-of-function version of UNC-9, and it may or may not represent UNC-9 localization in vivo.

We thank the reviewer for their comment. Due to the strong expression of UNC-9 in the body wall muscle cells, we do not think we could reliably visualize endogenous UNC-9 gap junction formed specifically between DD neurons. However, we believe that UNC-9::7×GFP represents the endogenous UNC-9 localization for the following reasons. First, in our previous work, we showed that UNC-9::GFP exhibits the same localization pattern as endogenous UNC-9 detected with the anti-UNC-9 antibody in the body wall muscles (Chen et al., Curr Biol,2007; Liu et al. JBC 2006). Second, in the revised manuscript, we examined the localization of N-terminally tagged GFP::UNC-9 which was shown to be functional (Meng et al., 2016). We observed that GFP::UNC-9 was localized at the axonal and dendritic tiling border of DD5 and DD6 similar to UNC-9::7×GFP. We therefore believe that UNC-9::7×GFP can be used to visualize the subcellular localization of UNC-9 even though it has a compromised channel activity. We have included the representative images of GFP::UNC-9 in Figure 2H.

2) Test whether unc-9 is also required for DD5/DD6 synapse tilling in other genetic backgrounds, in which DD5 and DD6 have overlapping axons. one concern is whether unc-9 is only required for synapse tilling when egl-20 is mutated.

From a candidate screening, we found that the *sax-2(ky216)* mutant exhibits a mild axonal tiling defect (see Author response image 1). We examined the presynaptic tiling defects in the *sax-2(ky216); unc-9(e101)* double mutants and found that there is a tendency that *unc-9(e101)* enhances the presynaptic tiling defect in *sax-2(ky216)* mutants, despite no statistically significant differences. This is likely due to the smaller degree of axonal tiling defect in *sax-2(ky216)* mutant compared with *egl-20(n585)* mutant, which limits the axonal region to form ectopic synapses compared with *egl-20(n585)* mutant.

**Author response image 1. sa2fig1:** 

3) DD5/DD6 don't have overlapping axons in wild type animals, and therefore all experiments for DD5/DD6 synapse tilling can only be carried out in mutants with abnormally overlapped DD5/DD6 axons. One concern is how "synapse tilling" in these abnormally developed axons represents synapse tilling during normal development. I understand it is challenging to address this in DD neurons, but it should be testable in other neurons who have overlapping axons and tilled synapses, for example DA neurons.

We examined presynaptic tiling between DA8 and DA9 in the *unc-9(e101)* mutant animals and did not observe presynaptic patterning defect. In DA9, UNC-9::7×GFP is predominantly localized at the distal dendrite. In the axon, we observed UNC-9::7×GFP puncta throughout the axon but we did not observe consistent localization in relation to their presynaptic domain (Figure 2—figure supplement 7).

Reviewer #3 (Recommendations for the authors):My major comment relates to the potential advance offered by the studies. The impact of the findings would be enhanced if the authors were able to define mechanisms for UNC-9 localization and/or function in tiling. Nonetheless, the findings of the study as presented are provocative and suggest a potentially novel mechanism for synaptic tiling through regulated expression/localization of gap junction proteins. This will be of interest to the field and sets the stage for follow up studies of mechanism.In several instances, the writing is unclear:Line 148: "we observed minimal overlaps between the axons and the dendrites of DD5 and DD6" would be more clearly stated as "minimal overlap between either axons or dendrites of…"Lines 168-9 are redundant with lines 173-4Lines 206-7: "the position of DD5 postsynaptic dendritic spine is determined by the length of DD5 dendrite". This is unclear, ACR-12::GFP puncta appear to extend along the full length of the dendrite.Lines 275-6: Delete "both".

We thank the reviewer for pointing these out, we have rectified the errors including the ones indicated above.